# THANOS: A BLOCK-WISE PRUNING ALGORITHM FOR EFFICIENT LARGE LANGUAGE MODEL COMPRESSION

## ABSTRACT

This paper presents Thanos, a novel weight-pruning algorithm designed to reduce the memory footprint and enhance the computational efficiency of large language models (LLMs) by removing redundant weights while maintaining accuracy. Thanos introduces a block-wise pruning strategy with adaptive masks that dynamically adjust to weight importance, enabling flexible sparsity patterns and structured formats, such as $n : m$ sparsity, optimized for hardware acceleration. Experimental evaluations demonstrate that Thanos achieves state-of-the-art performance in structured pruning and outperforms existing methods in unstructured pruning. By providing an efficient and adaptable approach to model compression, Thanos offers a practical solution for deploying large models in resource-constrained environments.

## 1 INTRODUCTION

Large Language Models (LLMs) (Brown, 2020; Achiam et al., 2023) have shown great performance in a variety of generative tasks (Bommarito et al., 2023; Wei et al., 2022; Bubeck et al., 2023). However, a key problem associated with its deployment is large memory footprint and high computational/inference complexity: to be able to train and run such models, a large number of powerful and expensive GPUs are needed. The cost of modern GPUs might be too large for individuals, which is why the largest and most advanced LLMs run on large computational centers. To make inference more affordable, model compression via quantization or pruning seems to be necessary.

Many recent papers advances have focused on quantization in neural network compression (Dettmers et al., 2022; Frantar et al., 2022; Xiao et al., 2023; Ahmadian et al., 2023; Malinovskii et al., 2024), where model parameters are converted to lower bit-level formats to reduce resource usage. Rapid progress in LLM quantization research has resulted in significant improvements in efficiency, and large reduction of computational cost (Sheng et al., 2023; Lin et al., 2024).

In this work, we revisit post training model pruning (LeCun et al., 1989; Hassibi & Stork, 1992; Han et al., 2015; Hubara et al., 2021a; Frantar & Alistarh, 2022; Kwon et al., 2022), as an alternative to quantization for reducing model size. Pruning works by eliminating certain weights, effectively setting them to zero while minimizing the impact on the quality of the model on downstream tasks. In particular, we present Thanos – a new and efficient LLM pruning method that does not require retraining nor training from scratch.

### 1.1 THE STRUCTURE OF LLMS

Large language models (LLMs) (Zhang et al., 2022; Touvron et al., 2023; Dubey et al., 2024) are composed of multiple sequential transformation blocks each of which contains several linear layers. These linear layers consist of weight matrices and bias vectors. In our work, we focus on pruning the weights captured in the linear layers only; we do so because they represent most of the parameters of a LLM (Frantar & Alistarh, 2022; Kwon et al., 2022).

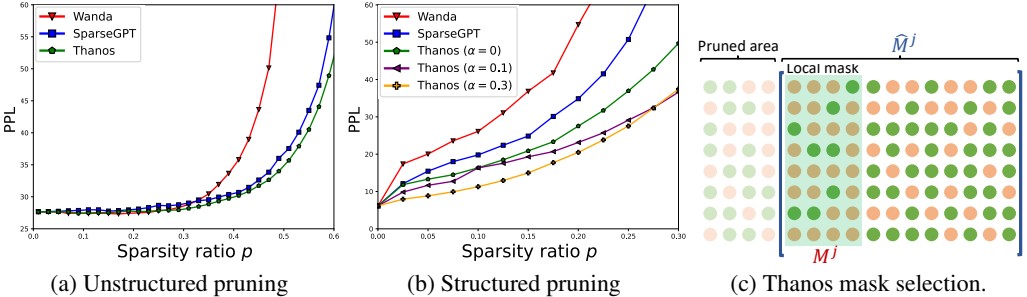

(a) Unstructured pruning     (b) Structured pruning     (c) Thanos mask selection.

Figure 1: (a, b): Evaluation of Wikitext2 perplexity pruned by different methods: Wanda, SparseGPT and Thanos (our). (a) Unstructured pruning (OPT-125M), (b) Structured pruning (LLaMA-3 8B). (c): Demonstration of Thanos mask selection algorithm. Entries of $M$ that are equal to one are represented by orange circles, while zeros are depicted with green circles.

## 1.2 POST TRAINING PRUNING

Post-training pruning refers to the practical scenario where a pre-trained and optimized model is provided, along with a small set of calibration data. The goal in this setting is to derive a compressed version of the weights, which could involve the introduction of sparsity via pruning, or the application of quantization, or both.

Although this approach initially gained popularity in quantization research (Hubara et al., 2021b; Nagel et al., 2020; Li et al., 2021), recent studies have also demonstrated the effectiveness of pruning (Hubara et al., 2021a; Frantar & Alistarh, 2022; Kwon et al., 2022).

We follow a common pruning paradigm used in data-aware methods (Frantar & Alistarh, 2023; Sun et al., 2023), where pruning is done sequentially, block by block. For each block, we first perform a forward pass to collect inputs for all linear layers, then prune each layer independently to a target sparsity. After pruning, a second forward pass through the block computes updated inputs for the next block. This process repeats across all blocks until the entire model is pruned (Frantar & Alistarh, 2022; Kwon et al., 2022).

The key novelty of our work is in *how* pruning is performed.

## 1.3 CONTRIBUTION OF THIS WORK

We now summarize the key contributions of our work.

1. We introduce *Thanos*, a block-wise pruning algorithm that efficiently updates weight blocks with minimal performance loss. A complexity comparison with other methods is shown in Table 4 (Section D).

2. We propose a new structured pruning technique using outlier row detection, achieving substantial improvements in perplexity and zero-shot performance over baselines.

3. Thanos supports the popular semi-structured $n{:}m$ format (Zhou et al., 2021; Hubara et al., 2021a; Mishra et al., 2021), enabling speedups on NVIDIA Ampere GPUs via its 2:4 implementation (Lin et al., 2023; NVIDIA Corporation, 2020).

4. Our evaluation across various language modeling tasks shows Thanos consistently outperforms existing structured pruning methods across sparsity levels (Fig. 1, Table 1).

5. We release our implementation of Thanos publicly to support future research in model compression and optimization.

## 1.4 NOTATION

All key notation used in this paper is summarized in a tabular form in Section A; see Table 2.

## 2 Formulation of the problem

### 2.1 Objective function

Post-training compression typically breaks down the task of compressing the entire model into smaller, layer-wise sub-problems. Recent studies (Frantar & Alistarh, 2023; Sun et al., 2023) have shown that post training pruning with limited data-based objectives can be very efficient. In such a setup we aim to minimize the difference between the output of linear layer before and after pruning (Hubara et al., 2021a).

Let us now consider a weight matrix $W \in \mathbb{R}^{c \times b}$ associated with a single layer, and the corresponding *input/calibration matrix* $X \in \mathbb{R}^{b \times a}$. By $\widehat{W} \in \mathbb{R}^{c \times b}$ we denote the pruned weight matrix. Let $M \in \{0,1\}^{c \times b}$ be a *pruning mask* – a matrix encoding the entries of $W$ that should be pruned. Every element $M_{ij}$ is either 0 or 1, with $M_{ij} = 1$ indicating that the weight $W_{ij}$ is to be eliminated. The *sparsity ratio* $p \in [0,1)$ associated with mask $M$ is the quantity $p := \frac{\|M\|_F^2}{cb}$, where $\|\cdot\|_F$ denotes the Frobenius norm. The larger $p$, the more information we lose.

The objective function $f(\widehat{W})$ can be written in the following way:

$$f(\widehat{W}) := \|(\widehat{W} - W)X\|_F^2. \tag{1}$$

With the aforementioned notations we introduce the following discrete constrained optimization problem of pruning a linear layer $W$:

$$\min_{\substack{\widehat{W} \in \mathbb{R}^{c \times b} \\ M \in \{0,1\}^{c \times b}}} \left[ \left\| \left( \widehat{W} - W \right) X \right\|_F^2 \right], \quad \text{subject to} \quad \begin{cases} \text{for all } ij, \text{ if } M_{ij} = 1, \text{ then } \widehat{W}_{ij} = 0, \\ \|M\|_F^2 = \lfloor pcb \rfloor. \end{cases} \tag{2}$$

where $\lfloor \cdot \rfloor : \mathbb{R} \to \mathbb{Z}$ is a floor function, defined by $\lfloor x \rfloor := \max \{n \in \mathbb{Z} \mid n \leq x\}$. We use the floor function in constraints of (2) because $\|M\|_F^2 \in \{0, \cdots, cb\}$ can only take integer values. Note that $\|M\|_F^2 \in \{0, \cdots, cb\}$ indicates how many parameters should be removed from $W \in \mathbb{R}^{c \times b}$.

The problem (2) is difficult due to its discrete sparsity constraints, making it NP-hard (Blumensath & Davies, 2008). As a result, exact solutions are infeasible for large layers, and most methods use approximations.

A common strategy splits the task into two steps: selecting a pruning mask and reconstructing the remaining weights (He et al., 2018; Kwon et al., 2022; Hubara et al., 2021a). The mask $M$ is chosen using some metric, such as weight magnitude (Zhu & Gupta, 2017), and the unpruned weights are then optimized. With a fixed mask, this reduces to a tractable least-squares problem.

### 2.2 Pruning of a single parameter of the layer

We will make the following simplification for the problem (2) and its constraints to be able to effectively solve the optimal pruning problem: let us solve this problem for only removing a single weight from $W$.

That is being said: our task is to solve the problem (2) with only one $M_{kq} = 1$ ($\widehat{W}_{kq} = 0$) for some $k \in \{1, \cdots, c\}$ and $q \in \{1, \cdots, b\}$. For all $i \neq k$ and $j \neq q$, $M_{ij} = 0$. Then the problem (2), can be written in the following form:

$$\min_{\Delta_{k:} \in \mathbb{R}^{1 \times b}} \left[ g(\Delta_{k:}) := \|\Delta_{k:} X\|_2^2 \right], \quad \text{subject to} \quad \Delta_{k:} e^q + W_{kq} = 0, \tag{3}$$

where $\Delta_{k:} = (\widehat{W} - W)_{k:} \in \mathbb{R}^{1 \times b}$ is the change of the $k^{\text{th}}$ row of the weight matrix $W$ and $e^q \in \mathbb{R}^{1 \times b}$ is a unit vector with 1 at the $q^{\text{th}}$ position and 0 everywhere else.

In practical scenarios, we prune layer $W$ to achieve a target sparsity level of $p$. So if we aim to remove multiple weights from the linear layer $W$, an effective approach is to apply the single-weight pruning solution of (3) iteratively. This approach removes the least important weights one by one until the desired sparsity is reached.

# 3 BACKGROUND AND RELATED WORK

## 3.1 MAGNITUDE PRUNING

One of the simplest and most intuitive methods for pruning LLMs is Magnitude pruning, first introduced by (Han et al., 2015), is a widely used method for introducing sparsity in neural networks. This technique eliminates individual weights by evaluating their magnitudes, discarding those that fall below a specific threshold. Additional information regarding the magnitude pruning technique is provided in the Appendix G.1.

For LLM, data-aware pruning methods which leverage small calibration datasets consistently yield superior results (He et al., 2018; Kwon et al., 2022; Hubara et al., 2021a; Frantar & Alistarh, 2023; Sun et al., 2023). For this reason, our study primarily focuses on data-aware techniques, as they strike a better balance between sparsity and performance, even with limited calibration data.

## 3.2 OBD AND OBS

A classical approach to solving the problem (3) was presented by Optimal Brain Damage (OBD) (Hassibi & Stork, 1992) and Optimal Brain Surgeon (OBS) (LeCun et al., 1989).

By establishing the solution for the problem (3), we obtain the following loss function $S_{kq}^{\text{OBS}}$ from removing a weight $W_{kq}$ and the optimal update rule $\Delta_{k:}^{\star} = \left(\widehat{W} - W\right)_{k:} \in \mathbb{R}^{1 \times b}$ for the row $W_{k:} \in \mathbb{R}^{1 \times b}$:

$$\Delta_{k:}^{\star} = -\frac{W_{kq}}{H_{qq}^{-1}} H_{q:}^{-1}, \qquad S_{kq}^{\text{OBS}} := \frac{1}{2} \frac{W_{kq}^2}{H_{qq}^{-1}}, \tag{4}$$

where $H = 2XX^{\top} \in \mathbb{R}^{b \times b}$ is the Hessian matrix of the function (3).

Optimal Brain Damage (Hassibi & Stork, 1992) assumed that the Hessian $H$ is a diagonal matrix, which makes it very simple to find its inverse. So the pruning metric from (4) simplifies to

$$S_{kq}^{\text{OBD}} = \frac{1}{2} \frac{W_{kq}^2}{H_{qq}^{-1}} = \frac{1}{2} \frac{W_{kq}^2}{(H_{qq})^{-1}} = W_{kq}^2 \sum_{i=1}^{a} X_{qi}^2 = W_{kq}^2 \|X_{q:}\|_2^2 = \left(|W_{kq}|\|X_{q:}\|_2\right)^2, \tag{5}$$

where $X_{q:} \in \mathbb{R}^{1 \times a}$ is the $q^{\text{th}}$ row of the matrix $X$.

Additional information regarding the OBD and OBS can be found in the Appendix G.2.

## 3.3 SPARSEGPT

SparseGPT (Frantar & Alistarh, 2023) addresses the parameter pruning challenge by sequentially pruning parameters from left to right. SparseGPT only considers parameters on the right side of the sequence, ensuring that the Hessian remains constant for all rows.

The algorithm iteratively traverses the weight matrix in blocks, employing the same procedure: updating the mask, adjusting the remaining weights, and refining the Hessian matrix. This iterative approach enables structured pruning of the model, ensuring that while numerous weights are eliminated, the performance loss is minimized by selecting which weights to prune by the metric $S_{kq}^{\text{OBS}}$ from (4) and adapting the remaining weights column by column by the update rule $\Delta_{k:}^{\star}$. Additional information regarding SparseGPT is provided in the Appendix G.3.

## 3.4 WANDA

Wanda (Sun et al., 2023) attempts to simplify the intensive computation of the inverse of the Hessian by assuming the Hessian to be diagonal. Consequently, the relationship between OBD and OBS is analogous to the relationship between Wanda and SparseGPT (Frantar & Alistarh, 2023). Therefore, the pruning metric (4) simplifies to (5). Additional information regarding Wanda is provided in the Appendix G.4.

# 4 THANOS PRUNING

SparseGPT (Frantar & Alistarh, 2023) prunes weights column by column, removing only one weight per row at a time. However, this approach fails to account for the cumulative impact of removing multiple columns simultaneously. Since the total change in weights cannot be approximated as the sum of independent changes from removing individual columns by (4), updating multiple weights at once can lead to more accurate adjustments and improved pruning performance.

## 4.1 UPDATING SEVERAL WEIGHTS AT A TIME

Assuming we aim to prune $s$ weights at a time, the problem becomes analogous to (3), but now includes $s$ constraints. Specifically, the weights $W_{kq_1}, \cdots, W_{kq_s}$ must be removed, where $1 \leq q_1 < \cdots < q_s \leq b$ represent the indices of the weights selected for pruning:

$$\min_{\Delta_{k:} \in \mathbb{R}^{1 \times b}} \left[ g(\Delta_{k:}) := \|\Delta_{k:} X\|_2^2 \right], \quad \text{s.t.} \quad \Delta_{k:} e^{q_1} + W_{kq_1} = 0, \; \cdots, \; \Delta_{k:} e^{q_s} + W_{kq_s} = 0. \quad (6)$$

Let us define matrices $R \in \mathbb{R}^{s \times b}$, $\widehat{R} \in \mathbb{R}^{s \times s}$ and a vector $u \in \mathbb{R}^{1 \times s}$ in the following way:

$$R := \begin{pmatrix} H_{q_1:}^{-1} \\ \vdots \\ H_{q_s:}^{-1} \end{pmatrix} \in \mathbb{R}^{s \times b} \qquad \begin{aligned} \widehat{R} &:= (R_{:q_1} \; \cdots \; R_{:q_s}) \in \mathbb{R}^{s \times s}, \\ u &:= (W_{kq_1} \; \cdots \; W_{kq_s}) \in \mathbb{R}^{1 \times s}. \end{aligned} \quad (7)$$

The solution of the problem (6) can be written in the following way:

$$\widehat{\Delta}_{k:} = -u\widehat{R}^{-1}R, \quad S_k := \tfrac{1}{2} u \widehat{R}^{-1} R H R^\top (\widehat{R}^{-1})^\top u^\top, \quad (8)$$

where $\widehat{\Delta}_{k:} \in \mathbb{R}^{1 \times b}$ is the optimal update rule for the $k^{\text{th}}$ row of $W \in \mathbb{R}^{c \times b}$ for the problem (6) and $S_k$ is the value of the loss function (6) with optimal update rule $\widehat{\Delta}_{k:}$. Additional details on the derivation of these expressions are provided in Appendix H.1.

The main challenge with this approach lies in its computational complexity. Selecting the optimal indices $q_1, \cdots, q_s$ involves a combinatorial explosion, with $\binom{b}{s} = \frac{b!}{s!(b-s)!}$ possible combinations per row. For each combination, computing $\widehat{\Delta}_{k:}$ and $S_k$ requires $\mathcal{O}\left( \frac{b!}{s!(b-s)!} \left( s^3 + c^2 + cs \right) \right)$ operations, making the method infeasible for LLMs.

The problem becomes much simpler if the pruning mask is predefined—for example, by selecting the $s$ smallest $S_{kq}^{\text{OBD}}$ values from the same row using the metric in (5). This removes the need to search for which weights to prune, reducing the task to two steps: selecting the mask and adjusting the retained weights.

## 4.2 PRUNING METRIC

Thanos employs the same pruning metric as Wanda. We showed that Wanda provides an optimal solution for removing a single weight at a time without adjusting the remaining weights (Appendix H.3).

## 4.3 PRUNING IN A BLOCK-WISE MANNER

Computing the optimal weight updates from Equation (8) for the full matrix $W \in \mathbb{R}^{c \times b}$ is computationally expensive due to the size of the resulting linear systems. To address this, we divide $W$ into smaller column-wise blocks of size $B$, making each subproblem tractable. This idea, also used in SparseGPT (Frantar & Alistarh, 2023), enables dynamic pruning mask updates, which is beneficial since weights can change significantly during pruning.

Let $B \in \{1, \ldots, b\}$ be the number of columns per block (distinct from $s$, the number of pruned weights per row). Then $W$ is split into $\lceil \frac{b}{B} \rceil$ local blocks, where the ceiling function is defined as $\lceil x \rceil := \min\{n \in \mathbb{Z} \mid n \geq x\}$. Smaller $B$ reduces the number of pruned weights per row, simplifying computation. The algorithm proceeds block by block, solving Equation (8) for each row and updating all remaining weights to the right of the current block.

---

**Algorithm 1** Thanos pruning algorithm (Unstructured)

---

1: **Initialization:** Input matrix $X \in \mathbb{R}^{b \times a}$, weight matrix $W \in \mathbb{R}^{c \times b}$, block size $B$, sparsity ratio $p \in [0, 1)$.
2: $r \leftarrow pcb, \quad H \leftarrow 2XX^\top$    *// Compute number of parameters to remove and the Hessian matrix*
3: **for** $j_1 = 1, \ B, \ 2B, \ \cdots, \ \lfloor \frac{b}{B} \rfloor B$ **do**
4:     $j_2 \leftarrow \min\{b, j_1 + B\}$
5:     $\widehat{M} \overset{(4.4)}{\leftarrow} \psi_X(W_{:,j_1:b}, r), \quad M \leftarrow \widehat{M}_{:,1:B}, \quad r \leftarrow r - \|M\|_F^2.$    *// compute global residual mask, local block mask and update number of parameters to remove*
6:     **for** $i = 1, \cdots, c$ **do**
7:        $w \leftarrow W_{i,j_1:b}, \quad q \overset{(4.5)}{\leftarrow} \phi(M_{i:}) \in \mathbb{N}^s$    *// select one row of weights and find indexes for removal*
8:        $R \overset{(7)}{\leftarrow} \left((H_{q_1:}^{-1})^\top \cdots (H_{q_s:}^{-1})^\top\right)^\top, \quad \widehat{R} \overset{(7)}{\leftarrow} (R_{:q_1} \cdots R_{:q_s}), \quad u \overset{(7)}{\leftarrow} (w_{q_1} \cdots w_{q_s})$
9:        $W_{i,j_1:b} \overset{(8)}{\leftarrow} w - u\widehat{R}^{-1}R$    *// update the selected row*
10:    **end for**
11:    $H \leftarrow 2(XX^\top)_{j_2:b,j_2:b}$    *// update the Hessian*
12: **end for**

---

### 4.4 UNSTRUCTURED SPARSITY

Thanos dynamically constructs the global residual mask $\widehat{M}^j$ based on the number of elements already pruned and the target sparsity level. Consequently, this algorithm is not restricted to local block-wise or row-wise sparsity patterns, enabling effective handling of weight adjustments throughout pruning. A visual representation of this process is provided in (Fig. 1c).

We define a mapping $\psi_X : \mathbb{R}^{c \times b} \to \{0, 1\}^{c \times b}$, which operates on a portion of the weight matrix $W$ and produces a pruning mask. For each element $W_{ij}$, this mapping calculates the pruning metric $|W_{ij}| \|X_{j:}\|_2$. It then returns a mask that contains ones at the positions corresponding to the $r$ smallest values of this metric:

$$\psi_X(W, r) := \text{mask of the } r \text{ weights } W_{ij} \text{ with the smallest } |W_{ij}| \|X_{j:}\|_2 \text{ values.}$$

See additional examples in Section G.4.

### 4.5 HOW TO GET INDICES OF WEIGHTS FOR REMOVAL

As discussed previously in Section 4.1, we require the indices $1 \le q_1 < \cdots < q_s \le b$ of the weights designated for removal. To facilitate this, we define a mapping $\phi : \{0, 1\}^B \to \mathbb{N}^s$ that transforms each vector $x \in \{0, 1\}^B$ into a vector of natural indices of its non-zero elements:

$$\phi(x) := \text{ vector of indices of non-zero elements from } x.$$

For instance, applying the mapping $\phi$ to the vector $x = (1, 0, 0, 1, 1)$ yields $\phi(x) = (1, 4, 5)$. Similarly, for $x = (0, 0, 1, 1, 0)$, we obtain $\phi(x) = (3, 4)$, and so forth. With this mapping, we can retrieve the indices for removal in each row of the mask $M$ as follows: $(q_1, \cdots, q_s) = \phi(M_{i:})$, where $M_{i:}$ represents elements from the $i^{\text{th}}$ row of the mask $M$. The mapping $\phi$ allows us to systematically identify the indices of the weights to be pruned within each specified row of the mask.

### 4.6 COMPLEXITY AND EFFICIENT IMPLEMENTATION

The comprehensive algorithm is presented in Algorithm 1. Table 4 presents a comparison of the computational complexities of all methods under consideration. Additional information on the efficient implementation of Thanos can be found in the Appendix I.1.

### 4.7 STRUCTURED SPARSITY

Structured sparsity can be more useful for practical usage because it does not require special cores for acceleration. In this type of sparsity, the entire column is removed, so the weight matrix $W$ can be made smaller without the need for additional storage of indices.

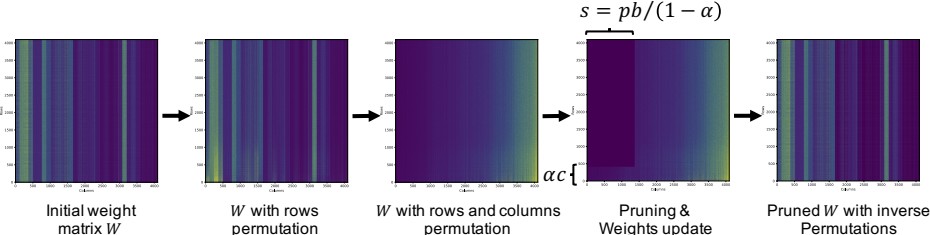

Figure 2: Demonstration of main steps of structured pruning by Thanos algorithm.

---

**Algorithm 2** Thanos pruning algorithm (Structured)

---

1: **Initialization:** Input matrix $X \in \mathbb{R}^{b \times a}$, weight matrix $W \in \mathbb{R}^{c \times b}$, percent of outlier rows $\alpha \in [0, 1)$, sparsity ratio $p \in [0, 1)$.
2: $s \leftarrow \lceil \frac{pb}{1-\alpha} \rceil$, $\quad H \leftarrow 2XX^\top$ $\quad$ *// Compute number of columns to remove and the Hessian matrix*
3: Compute $\{h_i\}_{i=1}^c$ by (10)
4: $Q \leftarrow Q(W, \{h_i\}_{i=1}^c)$, $\quad W' \leftarrow QW$ $\quad$ *// Permute rows (Section H.4.4)*
5: Compute $\{v_j\}_{j=1}^b$ by (11)
6: $P \leftarrow P(W'_{1:c-\lceil \alpha c \rceil}, \{v_j\}_{j=1}^b)$, $\quad W \leftarrow PW'$ $\quad$ *// Permute columns*
7: $W_{:,1:s} \overset{(8)}{\leftarrow} W_{:,1:s} - W_{:,1:s} \left( H_{1:s,1:s}^{-1} \right)^{-1} H_{1:s,:}^{-1}$ $\quad$ *// prune*
8: $W \leftarrow Q^\top W P^\top$ $\quad$ *// Perform the inverse permutations*

---

In case when we remove weights by eliminating the entire column, we do not need to solve the equation (8) for every single row since all indices $q_1, \cdots, q_s$ will be the same. For structured pruning, we can apply permutation of the columns of $W$ to make $q_1 = 1, \cdots, q_s = s$, so we only would need to remove the first $s$ columns of the weights matrix $W$ (More on permutations is in Section H.4.4). After pruning we need to perform the inverse permutation to get the correct weight matrix with all columns in the right order.

For structured sparsity, the equation (8) can be written as

$$\widehat{\Delta}_{k:} = -W_{:,1:s} \left( H_{1:s,1:s}^{-1} \right)^{-1} H_{1:s,:}^{-1}. \tag{9}$$

Due-to this simplification, Thanos is significantly faster than SparseGPT in structured pruning. More details are in Section I; see Fig 8. The Thanos pruning algorithm for structured sparsity is in (Alg. 2). The illustration of the overall pipeline is shown in Fig. 2.

### 4.7.1 OUTLIER ROWS (DEFINITION OF $\alpha$)

Structured pruning affects every row of $W$, but preserving certain rows may be beneficial. Prior work (Bondarenko et al., 2023; Sun et al., 2024; Yu et al., 2024) has shown that a small number of outlier weights play a crucial role in maintaining LLM inference quality. Motivated by this, we incorporate a selective pruning strategy that retains a fraction of rows during structured pruning.

Let $h_i, i \in \{1, \cdots, c\}$ to be the loss (2) induced by removal of the $i^{\text{th}}$ row of the weight matrix $W$, then $h_i$ can be written as

$$h_i := \|W_i X\|_2^2. \tag{10}$$

Let us denote the percent of outliers as $\alpha \in [0, 1)$, where $\alpha = 0$ means we prune all rows, $\alpha = 0.5$ means $50\%$ of rows are outlier-rows, we will not prune them, $\alpha = 1$ means that we perceive all rows as outliers, so nothing will be pruned. The number of outlier rows is $\lceil \alpha c \rceil$.

Note that if your goal is to achieve a fixed level of sparsity $p$, then you have to remove $s = \lceil \frac{pb}{1-\alpha} \rceil$ columns. As expected, with increasing $\alpha$, we have to prune more columns.

We choose columns for removal by computing the loss induced by every column and selecting those with the smallest value. Let $v_j, j \in \{1, \cdots, b\}$ to be the partial loss (2) induced by removal of the $j^{\text{th}}$ column of the weight matrix $W$, then $v_j$ can be written as

$$v_j := \|W_{1:c-\lceil \alpha c \rceil, j} \otimes X_j\|_F^2, \tag{11}$$

Table 1: Comparison of WikiText Perplexity (left) and Average Zero-Shot Accuracy (right) of pruned LLaMA-2 and LLaMA-3 models. More results are in Section E.

| | | Perplexity (↓) – WikiText | | | | | | | | Zero-Shot Accuracy (↑) | | | | | | | |
| | | LLaMA-2 | | | | LLaMA-3 | | | | LLaMA-2 | | | | LLaMA-3 | | | |
| Method | Sparsity | 1.1B | 7B | 13B | 70B | 1B | 3B | 8B | 70B | 1.1B | 7B | 13B | 70B | 1B | 3B | 8B | 70B |
|---|---|---|---|---|---|---|---|---|---|---|---|---|---|---|---|---|---|
| Dense | 0% | 7.97 | 5.47 | 4.88 | 3.32 | 9.75 | 7.81 | 6.14 | 2.85 | 51.25 | 61.86 | 64.94 | 67.72 | 52.88 | 60.45 | 65.63 | 71.40 |
| Magnitude | | 24.29 | 16.03 | 6.83 | 5.36 | 1361.97 | 139.41 | 205.47 | 19.62 | 43.15 | 53.23 | 55.74 | 62.50 | 32.95 | 37.77 | 39.90 | 42.65 |
| SparseGPT | Unstruct. | 11.12 | 7.02 | 6.02 | 4.25 | **18.82** | 12.33 | 9.36 | **5.78** | **47.74** | 59.09 | 62.27 | 67.15 | 46.79 | 54.71 | **61.21** | **69.17** |
| Wanda | 50% | 11.50 | **6.92** | **5.97** | **4.22** | 23.38 | 13.01 | 9.83 | 5.82 | 46.85 | **59.16** | **62.62** | 66.75 | 44.72 | 53.28 | 59.48 | 68.64 |
| Thanos | | **11.05** | 6.96 | 6.03 | 4.23 | 19.54 | **12.28** | **9.32** | **5.78** | 47.50 | 58.91 | 62.06 | **67.64** | **46.86** | **54.78** | 60.48 | 69.11 |
| SparseGPT | | 63.26 | 42.10 | 38.72 | 15.56 | 248.44 | 93.52 | 82.19 | 36.75 | 36.87 | 37.20 | 38.90 | 52.00 | 35.80 | 37.38 | 38.97 | 47.12 |
| Wanda | Struct. | 170.25 | 83.12 | 75.15 | 468.96 | 1136.78 | 308.22 | 167.57 | 36.47 | 35.39 | 36.44 | 36.83 | 42.25 | 33.26 | 36.10 | 37.34 | 43.09 |
| Thanos ($\alpha = 0$) | 30% | 54.89 | 30.95 | 22.28 | 14.57 | 122.32 | 68.54 | 64.43 | 23.75 | 36.88 | **40.34** | 42.19 | 53.00 | 37.02 | **38.07** | 40.72 | 51.97 |
| **Thanos** ($\alpha = 0.1$) | | **44.32** | **28.16** | **21.32** | **9.82** | **107.61** | **57.57** | **36.61** | **16.62** | **37.57** | 39.37 | **43.46** | **57.58** | **37.62** | 37.64 | **41.47** | **55.26** |
| Magnitude | | 24.02 | 15.91 | 7.32 | 5.88 | 843.31 | 142.31 | 181.47 | 11.45 | 42.77 | 53.48 | 55.15 | 61.84 | 33.07 | 41.10 | 42.63 | 56.92 |
| SparseGPT | | 14.28 | 8.49 | 7.02 | 4.91 | 24.54 | 16.12 | 12.13 | 7.20 | 44.85 | 55.95 | 60.40 | 65.63 | 44.39 | 51.20 | 56.15 | 67.67 |
| Wanda | 4:8 | 16.79 | 8.60 | 7.01 | 4.77 | 44.75 | 21.17 | 14.51 | 7.10 | 43.58 | 55.53 | 60.13 | 65.52 | 40.39 | 48.37 | 54.04 | 65.98 |
| Thanos ($\alpha = 0$) | | 13.85 | 8.41 | 6.99 | 4.90 | 23.24 | 16.03 | 12.17 | 7.19 | **45.43** | 56.23 | 60.84 | 65.67 | 44.58 | 51.54 | 56.89 | 67.63 |
| **Thanos** ($\alpha = 0.1$) | | **12.82** | **7.86** | **6.60** | **4.51** | **20.38** | **14.46** | **10.97** | **6.29** | 45.36 | **57.11** | **61.05** | **66.58** | **44.89** | **52.28** | **57.86** | **68.76** |
| Magnitude | | 60.63 | 37.76 | 8.89 | 6.76 | 3288.83 | 387.64 | 2401.32 | 20.99 | 37.33 | 50.02 | 52.38 | 61.79 | 32.83 | 35.59 | 37.33 | 53.62 |
| SparseGPT | | 19.19 | 10.82 | 8.84 | 5.69 | 32.73 | 21.75 | 16.17 | 9.44 | 43.06 | 52.54 | 57.86 | 63.91 | **42.63** | 48.55 | 52.75 | 64.61 |
| Wanda | 2:4 | 27.15 | 12.13 | 8.98 | 5.48 | 78.58 | 35.14 | 24.15 | 9.22 | 40.91 | 51.35 | 56.11 | 63.97 | 37.63 | 44.46 | 47.40 | 62.84 |
| Thanos ($\alpha = 0$) | | 18.61 | 10.81 | 8.62 | 5.71 | 30.93 | 21.17 | 16.06 | 9.52 | **43.18** | 52.94 | 57.41 | 63.99 | 42.06 | 48.52 | 52.71 | 64.37 |
| **Thanos** ($\alpha = 0.1$) | | **16.31** | **9.68** | **7.83** | **4.98** | **26.24** | **18.72** | **13.92** | **7.39** | 42.81 | **54.53** | **58.93** | **65.65** | 42.52 | **49.88** | **54.39** | **66.36** |

where $\otimes$ denotes the outer product of two vectors.

In total, we perform two permutations – vertical (permute rows to move outliers in the end) and horizontal (permute columns to move weights for removal in the beginning). After the pruning procedure, we have to apply the inverse permutations to obtain the correct weight matrix with all columns and rows in the right order.

To facilitate structured pruning while preserving row-wise and column-wise order after pruning, we employ permutation matrices to reorder the rows and columns of the weight matrix $W$. More details on permutation matrices can be found in the Section H.4.4.

### 4.8 SEMI-STRUCTURED $n : m$ SPARSITY

The Thanos algorithm naturally extends to semi-structured patterns like the popular $n : m$ sparsity format (Zhou et al., 2021; Hubara et al., 2021a; Mishra et al., 2021), where each group of $m$ weights contains exactly $n$ zeros. This format offers speedups on hardware such as NVIDIA Ampere GPUs with optimized 2:4 support (Lin et al., 2023; NVIDIA Corporation, 2020). The structured variant of Thanos is detailed in Appendix H.4.5.

Thanos becomes more efficient in this setting (Alg. 8) since each row removes a fixed number of weights, enabling streamlined pruning and faster batched matrix operations. It outperforms SparseGPT for models up to 3B parameters (Appendix I).

## 5 EXPERIMENTS

### 5.1 MODELS AND EVALUATION.

We evaluate Thanos on two of the most widely used LLM model families: OPT family models (Zhang et al., 2022), and the LLaMA family models, including TinyLlama 1.3B (Zhang et al., 2024) LLaMA-2 (7B/13B/70B) (Touvron et al., 2023) and LLaMA-3 (1B/3B/8B/70B) (Dubey et al., 2024). Model performance is assessed across two key dimensions: zero-shot tasks and language modeling. For the zero-shot evaluation, we utilize seven tasks from the EleutherAI LM Harness (Gao et al., 2021) on a diverse set of zero-shot tasks like WinoGrande (Sakaguchi et al., 2021), OBQA (Mihaylov et al., 2018), BoolQ (Clark et al., 2019) PiQA (Tata & Patel, 2003), HellaSwag (Zellers et al., 2019), ARC-easy and ARC-challenge (Clark et al., 2018).

Consistent with prior research on LLM compression (Xiao et al., 2023; Frantar & Alistarh, 2023; Sun et al., 2023), we measure perplexity on the WikiText-2 (Merity et al., 2016) validation set.

Thanos is compared against three existing pruning techniques: Magnitude pruning (Han et al., 2015), SparseGPT (Frantar & Alistarh, 2023), and Wanda (Sun et al., 2023).

Since Thanos, SparseGPT, and Wanda require calibration data to compute input statistics, we ensure fair comparisons by using the same set of calibration data across all methods. This dataset consists of 128 sequences with a context length sampled from the C4 training set (Raffel et al., 2020).

In Thanos, we used a block size of $B = 128$ for unstructured sparsity. This relatively small block size was chosen because we observed a minimal impact on perplexity with varying $B$ values in the unstructured sparsity setting. In contrast, for structured sparsity patterns like 4:8 and 2:4, we opted for a larger block size of $B = 512$. Experiments with different blocksize are presented in Appendix C.

Unless stated otherwise, we set $\alpha = 0.1$ (i.e., $10\%$ of outlier rows). In semi-structured sparsity with $\alpha = 0.1$, the total sparsity $p$ decreases from $p = 0.5$ to $p = 0.45$. However, in structured sparsity, $p$ remains unchanged because, as $\alpha$ increases, additional columns are pruned: $s = \lceil \frac{pb}{1-\alpha} \rceil$.

## 5.2 PERPLEXITY EVALUATION

In terms of WikiText perplexity (Table 1), which evaluates a model's capability in language modeling (where lower values indicate better performance), Thanos demonstrates competitive results and, in some cases, achieves the best performance in unstructured sparsity. Specifically, Thanos outperforms existing methods in pruning TinyLlama 1.3B and all LLaMA-3 models ranging from 3B to 70B parameters, yielding lower perplexity scores than SparseGPT, Wanda, and Magnitude pruning.

For structured pruning, Thanos consistently surpasses other widely used pruning methods by a significant margin. Furthermore, it exhibits higher efficiency than SparseGPT (Section I), underscoring its advantages in structured pruning, particularly at high sparsity levels.

The superior performance of Thanos in structured sparsity stems from the fact that, when applying structured sparsity, the cumulative change in weights due to the removal of multiple columns (8) cannot be accurately approximated by the sum of individual changes resulting from the removal of each independent column.

## 5.3 ZERO-SHOT EVALUATION ON DIFFERENT LLM'S TASKS

For zero-shot evaluation (Table 1), which assesses model performance on unseen tasks (where higher values indicate better performance), Thanos remains competitive and outperforms other methods in unstructured sparsity. Notably, Thanos achieves the best zero-shot results for LLaMA-2 70B and LLaMA-3 models ranging from 1B to 3B parameters.

Similar to the perplexity results, Thanos consistently outperforms competing methods in structured and semi-structured sparsity. This demonstrates its strong generalization across various zero-shot tasks and its superior performance in structured pruning. A comprehensive set of results for different evaluation tasks across various sparsity structures is provided in Appendix E.

## 5.4 EXPERIMENTS WITH DIFFERENT BLOCKSIZES

We conducted several experiments using different block sizes $B \in \{8, \ldots, 4096\}$ to prune TinyLlama-1.1B (Zhang et al., 2024) in the Table 3. For unstructured sparsity, we observed that perplexity remained relatively stable across different block sizes. Based on these findings, we selected $B = 128$ for most experiments involving unstructured sparsity to balance efficiency and performance. However, for structured sparsity configurations, such as 4:8 or 2:4, we found that pruning was more effective with larger block sizes.

## 6 CONCLUSION AND LIMITATIONS

Concluding comments and limitations of our approach can be found in Section B.

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

## A  TABLE OF FREQUENTLY USED NOTATION

Table 2: Notation table

| | | |
|---|---|---|
| $i, j$ | – | Matrix indexes for rows and columns respectively. |
| $k, q$ | – | Matrix indexes for row and column for pruning respectively. |
| $x_j$ | – | $j^{\text{th}}$ element of the vector $x$. |
| $e^j$ | – | Unit vector with 1 at the $j^{\text{th}}$ position and 0 everywhere else. $e^j \in \mathbb{R}^{b \times 1}$. |
| $W_{ij}$ | – | one entry of the matrix $W$ on the intersection of the $i^{\text{th}}$ row and $j^{\text{th}}$ column. |
| $W_{i:}$ | – | $i^{\text{th}}$ row of the matrix $W$, $W_{i:}$ is a row-vector. |
| $W_{:j}$ | – | $j^{\text{th}}$ column of the matrix $W$, $W_{:j}$ is a column-vector. |
| $W_{i,j_1:j_2}$ | $:=$ | $(W_{ij_1} \cdots W_{ij_2}) \in \mathbb{R}^{1 \times (j_2 - j_1)}, \quad 1 \le j_1 \le j_2 \le b.$ |
| $W_{i_1:i_2,j}$ | $:=$ | $(W_{i_1 j} \cdots W_{i_2 j})^\top \in \mathbb{R}^{(i_2 - i_1) \times 1}, \quad 1 \le i_1 \le i_2 \le c.$ |
| $W_{i_1:i_2,j_1:j_2}$ | $:=$ | $(W_{i_1:i_2,j_1} \cdots W_{i_1:i_2,j_2}) \in \mathbb{R}^{(i_2 - i_1) \times (j_2 - j_1)}, \quad 1 \le j_1 \le j_2 \le b, \quad 1 \le i_1 \le i_2 \le c.$ |
| $W_{i:,j:}$ | $:=$ | $(W_{i:c,j} \cdots W_{i:c,b}) \in \mathbb{R}^{(c-i) \times (b-j)}, \quad 1 \le j \le b, \quad 1 \le i \le c.$ |
| $W$ | – | Linear layer (matrix) that we aim to prune, $W \in \mathbb{R}^{c \times b}$. |
| $X$ | – | Input matrix for a linear layer $W$, $X \in \mathbb{R}^{b \times a}$. |
| $\widehat{W}$ | – | Sparse linear layer after pruning some weights from $W$, $\widehat{W} \in \mathbb{R}^{c \times b}$. |
| $\widetilde{W}$ | – | Modified linear layer, $\widetilde{W} \in \mathbb{R}^{c \times b}$. |
| $w$ | $:=$ | $W_{k:} \in \mathbb{R}^{1 \times b}$ – row of $W$ for pruning. |
| $v_j$ | – | Loss induced by removal of the $j^{\text{th}}$ column. |
| $h_i$ | – | Loss induced by removal of the $i^{\text{th}}$ row. |
| $\Delta$ | $:=$ | $\widehat{W} - W$ – change of the weight matrix (before and after pruning), $\Delta \in \mathbb{R}^{c \times b}$. |
| $\delta$ | $:=$ | $\Delta_{k:} \in \mathbb{R}^{1 \times b}$. |
| $p$ | $:=$ | $\frac{1}{cb}\|M\|_F^2$ – sparsity ratio $p \in [0, 1)$. |
| $B$ | – | Block size for parameters update, $B \in \{1, \cdots, b\}$. |
| $\widehat{f}(\widehat{W})$ | $:=$ | $\|(\widehat{W} - W)X\|_F^2$ – objective function that we aim to minimize while pruning. |
| $\widetilde{f}(\widetilde{W}, M)$ | $:=$ | $\left\|\left((1 - M) \odot \widetilde{W} - W\right)X\right\|_F^2$ – objective function that we aim to minimize while pruning. |
| $f(\Delta)$ | $:=$ | $\|\Delta X\|_F^2$ – objective function that we aim to minimize while pruning. |
| $g(\delta)$ | $:=$ | $\|\delta X\|_2^2$ – part of the objective function $f$, that depends only on one row of $W$. |
| $d$ | – | Number of calibration samples |
| $\delta^\star$ | – | Optimal change of one row of weights for the problem (35), $\delta^\star \in \mathbb{R}^{1 \times b}$. |
| $\Delta^\star$ | – | Optimal change of weights for the problem (35), $\Delta^\star \in \mathbb{R}^{c \times b}$. |
| $\widehat{\delta}$ | – | Optimal change of one row of weights for the problem (47), $\widehat{\delta} \in \mathbb{R}^{1 \times b}$. |
| $\widehat{\Delta}$ | – | Optimal change of weights for the problem (47), $\widehat{\Delta} \in \mathbb{R}^{c \times b}$. |
| $X^l$ | – | $l^{\text{th}}$ calibration sample, $X^l \in \mathbb{R}^{b \times a}$. |
| $F(\Delta)$ | $:=$ | $\frac{1}{d}\sum_{l=1}^d \|\Delta X^l\|_F^2$ – objective function in case of multiple calibration inputs. |
| $H$ | – | Hessian matrix for the function $g(\delta)$, $H = 2XX^\top$, $H \in \mathbb{R}^{b \times b}$. |
| $1 \in \mathbb{R}^{c \times b}$ | – | The matrix with size $c \times b$, all entries of which are equal to 1. |
| $0 \in \mathbb{R}^{c \times b}$ | – | The matrix with size $c \times b$, all entries of which are equal to 0. |
| $\|x\|_2$ | $:=$ | $\sqrt{\sum_{i=1} |x_i|^2}$ – $l^2$-norm of vector. |
| $\|A\|_F$ | $:=$ | $\sqrt{\sum_{i=1}\sum_{j=1} |A_{ij}|^2}$ – Frobenius norm of the matrix. |
| $\psi_X(W, r)$ | $:=$ | mask of the $r$ weights $W_{ij}$ with the smallest $|W_{ij}|\|X_{j:}\|_2$ values. |
| $\phi(h)$ | $:=$ | vector of indices of non-zero elements from $h$. |
| $M$ | – | Pruning mask. Equal to one only for entries of $W$ that should be removed after pruning. |
| $\widehat{M}$ | – | Global residual mask. Similar to $M$, but only the first $B$ columns are used for weights removal. |
| $\odot$ | – | Element-wise product of two matrices (Hadamard product). |
| $\otimes$ | – | Outer product of two vectors. |
| $\lfloor x \rfloor$ | $:=$ | $\max\{n \in \mathbb{Z} \mid n \le x\}$ – floor function. |
| $\lceil x \rceil$ | $:=$ | $\min\{n \in \mathbb{Z} \mid n \ge x\}$ – ceiling function. |

## B CONCLUSION & LIMITATIONS

### B.1 CONCLUSION

We introduced a novel pruning algorithm, Thanos, designed to perform block-wise weight pruning with minimal impact on model performance. Our contributions include the development of an adaptive pruning mask strategy that allows dynamic adjustments during pruning, thus enhancing the algorithm's flexibility beyond local block or row sparsity constraints. This adaptability enables Thanos to retain model generalization capabilities effectively, even as sparsity increases.

Our comprehensive evaluation demonstrates the competitive advantages of Thanos across various language modeling tasks and model sizes. Thanos significantly outperforms other approaches in structured and semi-structured sparsity patterns. In addition, Thanos maintains competitive performance while offering speed and efficiency improvements over SparseGPT.

A key reason for Thanos' superior performance in structured sparsity is that the cumulative change in weights from the removal of multiple columns (8) cannot be accurately approximated as the sum of independent changes from removing individual rows (4). Unlike SparseGPT, which prunes weights column by column and removes only one weight per row at a time, Thanos leverages this insight to update multiple weights simultaneously, leading to more accurate adjustments and improved pruning performance.

Furthermore, we introduced a novel approach to structured pruning that incorporates the identification of outlier rows. Our method demonstrates a significant enhancement in perplexity and zero-shot evaluation performance compared to existing baselines. Our open-source implementation of Thanos provides a valuable tool for researchers and practitioners, enabling them to leverage our method in their own work and fostering further advancements in model compression.

Overall, Thanos offers a powerful and flexible approach to model pruning, proving to be both computationally efficient and performance-preserving across a wide range of model sizes and sparsity levels. The adaptability of the Thanos pruning mechanism suggests its potential for broader applications in LLM compression.

### B.2 LIMITATIONS

A key limitation of Thanos is its computational complexity in unstructured and semi-structured sparsity. Unlike simpler magnitude-based pruning methods, Thanos requires solving multiple systems of linear equations to prune each block of weights. This significantly increases the computational cost, particularly with larger matrices or when targeting high sparsity levels. As model sizes and the number of parameters grow, the time and resources needed to perform these calculations become a limiting factor.

Moreover, while Thanos performs well across various model families, SparseGPT often surpasses it in zero-shot evaluation tasks in unstructured sparsity. This suggests that Thanos may face scalability challenges when applied to very large models or when attempting to achieve fine-grained sparsity patterns, as its adaptable weight updates do not always translate to optimal performance in these configurations.

Additionally, Wanda, generally not as competitive as Thanos and SparseGPT at lower sparsity levels, maintains robustness in unstructured sparsity for LLaMA-2 and LLaMA-3 models. This further highlights the importance of selecting a pruning method tailored to the specific use case and model architecture, as each approach exhibits unique strengths and limitations in different settings.

## C EXPERIMENTS WITH DIFFERENT BLOCKSIZE $B$

We conducted several experiments using different block sizes $B \in \{8, \ldots, 4096\}$ to prune TinyLlama-1.1B (Zhang et al., 2024) in the Table 3. For unstructured sparsity, we observed that perplexity remained relatively stable across different block sizes. Based on these findings, we selected $B = 128$ for most experiments involving unstructured sparsity to balance efficiency and performance. However, for structured sparsity configurations, such as 4:8 or 2:4, we found that pruning was more effective with larger block sizes.

Table 3: Pruning of TinyLlama-1.1B (Zhang et al., 2024) by Thanos with different blocksizes $B$.

| Blocksize $B$ | 8 | 64 | 128 | 256 | 1024 | 2048 | 4096 |
|---|---|---|---|---|---|---|---|
| Perplexity on Thanos 50% | 11.00 | 11.00 | 11.03 | 11.03 | 10.99 | 10.99 | 10.99 |
| Perplexity on Thanos 4:8 | 14.44 | 14.33 | 14.26 | 14.22 | 14.17 | 13.83 | 13.85 |
| Perplexity on Thanos 2:4 | 19.83 | 19.88 | 19.68 | 19.37 | 18.97 | 18.60 | 18.61 |

## D  COMPARISON OF COMPLEXITIES WITH OTHER METHODS

| Method | Complexity | Optimal Block Updates | Weights Update | Calibration Data |
|---|---|---|---|---|
| Magnitude (Han et al., 2015) | $\mathcal{O}(c^2 \log c)$ | ✗ | ✗ | ✗ |
| Wanda (Sun et al., 2023) | $\mathcal{O}(c^2 \log c)$ | ✗ | ✗ | ✓ |
| SparseGPT (Frantar & Alistarh, 2023) | $\mathcal{O}(c^3)$ | ✗ | ✓ | ✓ |
| Thanos (Unstructured & Semi-structured) NEW (Alg. 1, Alg. 8) | $\mathcal{O}(c^4/B + c^2 B^2)$ | ✓ | ✓ | ✓ |
| Thanos (Structured) NEW (Alg. 2) | $\mathcal{O}(c^3)$ | ✓ | ✓ | ✓ |

Table 4: Comparison of complexities between existing and our method. We assume linear layers $W \in \mathbb{R}^{c \times b}$ with $c = b$, and ignore sequence length $a$. $B \in \{1, \cdots, b\}$ is the block size used in Thanos.

## E  FULL ZERO-SHOT EVALUATION ON DIFFERENT LLM'S TASKS

### E.1  UNSTRUCTURED 50% SPARSITY

Table 5: Zero-Shot evaluation with different methods for OPT. Unstructured 50% sparsity.

| Params | Method | ArcC | ArcE | BoolQ | HellaSwag | OBQA | PiQA | WinoGrande | Average |
|---|---|---|---|---|---|---|---|---|---|
| | Dense | 50.28 | 16.60 | 55.44 | 62.95 | 29.17 | 43.52 | 19.03 | 39.57 |
| 125M | Magnitude | **52.64** | 14.20 | 60.52 | 57.45 | 27.30 | 33.08 | **19.97** | 37.88 |
| | SparseGPT | 52.41 | **15.80** | 61.90 | **62.02** | 27.90 | 40.32 | 19.20 | 39.94 |
| | Wanda | 51.85 | 14.00 | **62.08** | 61.32 | 28.19 | 39.90 | 19.20 | 39.51 |
| | **Thanos** | 51.85 | 15.60 | 61.16 | **62.02** | **28.27** | **41.33** | 19.62 | **39.98** |
| | Dense | 52.64 | 17.60 | 57.61 | 64.53 | 32.03 | 44.11 | 20.82 | 41.33 |
| 350M | Magnitude | **52.80** | 12.80 | 43.36 | 59.09 | 28.62 | 36.15 | 18.52 | 35.91 |
| | SparseGPT | 50.99 | 15.00 | 59.17 | **61.59** | **30.06** | **40.15** | 19.37 | 39.48 |
| | Wanda | 51.46 | 13.80 | 49.57 | 60.88 | 29.55 | 39.60 | 18.69 | 37.65 |
| | **Thanos** | 52.09 | **16.00** | **59.33** | 61.37 | 29.98 | 38.64 | **19.88** | **39.61** |

Table 6: Zero-Shot evaluation with different methods for LLaMA-2. Unstructured 50% sparsity.

| Params | Method | ArcC | ArcE | BoolQ | HellaSwag | OBQA | PiQA | WinoGrande | Average |
|--------|--------|------|------|-------|-----------|------|------|------------|---------|
| | Dense | 60.14 | 25.20 | 61.22 | 74.54 | 46.57 | 61.24 | 29.86 | 51.25 |
| 1.1B | Magnitude | 55.33 | 18.80 | 54.59 | 67.41 | 36.46 | 47.05 | 22.44 | 43.15 |
| | **SparseGPT** | **59.04** | 23.20 | 62.48 | **70.51** | **40.21** | **53.83** | **24.91** | **47.74** |
| | Wanda | 57.54 | 21.20 | **63.82** | 69.48 | 39.22 | 51.94 | 24.74 | 46.85 |
| | Thanos | 58.48 | **23.60** | 62.60 | 70.29 | 40.18 | 53.37 | 23.98 | 47.50 |
| | Dense | 69.06 | 31.40 | 77.68 | 78.07 | 57.15 | 76.35 | 43.34 | 61.86 |
| 7B | Magnitude | 63.30 | 26.80 | 62.97 | 71.60 | 49.13 | 64.06 | 34.73 | 53.23 |
| | SparseGPT | 69.77 | 29.20 | **75.99** | 75.68 | 52.74 | 71.93 | 38.31 | 59.09 |
| | **Wanda** | 67.17 | **30.60** | 75.32 | **76.66** | 52.68 | **72.31** | **39.42** | **59.16** |
| | Thanos | 68.59 | 28.40 | 75.50 | 76.22 | **53.04** | 71.89 | 38.74 | 58.91 |
| | Dense | 71.98 | 35.20 | 80.55 | 79.00 | 60.05 | 79.34 | 48.46 | 64.94 |
| 13B | Magnitude | 65.43 | 27.60 | 57.68 | 76.33 | 54.40 | 70.54 | 38.23 | 55.74 |
| | SparseGPT | 71.35 | **32.20** | **81.41** | 77.26 | 55.87 | 75.13 | 42.66 | 62.27 |
| | **Wanda** | **71.51** | 31.00 | 81.04 | **78.51** | **56.95** | **76.30** | **43.00** | **62.62** |
| | Thanos | **71.51** | 30.80 | **81.41** | 77.75 | 56.06 | 74.75 | 42.15 | 62.06 |
| | Dense | 77.90 | 37.20 | 84.00 | 82.21 | 55.35 | 82.95 | 54.44 | 67.72 |
| 70B | Magnitude | 73.64 | 35.40 | 71.60 | 78.84 | 52.60 | 76.20 | 49.23 | 62.50 |
| | SparseGPT | 77.82 | 37.00 | 85.05 | **81.39** | 54.35 | 81.55 | 52.90 | 67.15 |
| | Wanda | 77.58 | 37.20 | 83.40 | 81.07 | **54.50** | 81.30 | 52.22 | 66.75 |
| | **Thanos** | **79.01** | **37.40** | **85.30** | 81.12 | 54.45 | **82.25** | **53.92** | **67.64** |

Table 7: Zero-Shot evaluation with different methods for LLaMA-3. Unstructured 50% sparsity.

| Params | Method | ArcC | ArcE | BoolQ | HellaSwag | OBQA | PiQA | WinoGrande | Average |
|--------|--------|------|------|-------|-----------|------|------|------------|---------|
| | Dense | 60.62 | 26.40 | 64.04 | 74.48 | 47.73 | 65.57 | 31.31 | 52.88 |
| 1B | Magnitude | 51.07 | 12.80 | 37.86 | 54.62 | 26.72 | 28.62 | 18.94 | 32.95 |
| | SparseGPT | 55.56 | 20.60 | **63.03** | 67.90 | **39.29** | 54.42 | 26.71 | 46.79 |
| | Wanda | 55.80 | 18.00 | 61.62 | 65.23 | 35.18 | 52.44 | 24.74 | 44.72 |
| | **Thanos** | **55.96** | **22.60** | 62.45 | **68.17** | 38.61 | 53.45 | **26.79** | **46.86** |
| | Dense | 69.93 | 31.20 | 73.21 | 76.66 | 55.26 | 74.58 | 42.32 | 60.45 |
| 3B | Magnitude | 53.35 | 14.40 | 41.99 | 60.34 | 30.92 | 40.61 | 22.78 | 37.77 |
| | SparseGPT | 66.85 | 25.40 | **72.20** | 72.69 | **47.26** | **65.49** | 33.11 | 54.71 |
| | Wanda | 63.93 | 25.20 | 68.26 | 72.69 | 45.30 | 65.24 | 32.34 | 53.28 |
| | **Thanos** | **67.17** | **26.80** | 70.98 | **73.29** | 46.92 | 65.03 | **33.28** | **54.78** |
| | Dense | 72.77 | 34.80 | 81.31 | 79.71 | 60.19 | 80.13 | 50.51 | 65.63 |
| 8B | Magnitude | 52.96 | 22.00 | 42.91 | 59.96 | 29.85 | 46.72 | 24.91 | 39.90 |
| | **SparseGPT** | **72.14** | **30.20** | **79.48** | **76.66** | **53.87** | **73.65** | **42.49** | **61.21** |
| | Wanda | 70.56 | 29.20 | 77.89 | 75.90 | 51.20 | 71.34 | 40.27 | 59.48 |
| | Thanos | 71.19 | 29.60 | 78.26 | 76.17 | 53.51 | 73.23 | 41.38 | 60.48 |
| | Dense | 80.43 | 38.20 | 85.23 | 82.43 | 66.35 | 86.91 | 60.24 | 71.40 |
| 70B | Magnitude | 61.17 | 23.80 | 63.49 | 56.91 | 38.97 | 33.25 | 20.99 | 42.65 |
| | **SparseGPT** | **80.74** | 34.80 | **86.09** | 81.28 | 62.55 | 82.79 | **55.97** | **69.17** |
| | Wanda | 76.87 | 35.00 | 85.20 | **81.94** | 62.73 | **83.29** | 55.46 | 68.64 |
| | Thanos | 79.72 | **36.20** | 85.60 | 81.45 | **62.74** | 83.21 | 54.86 | 69.11 |

## E.2 Structured 30% sparsity

Table 8: Zero-Shot evaluation with different methods for OPT. Structured 30% sparsity.

| Params | Method | ArcC | ArcE | BoolQ | HellaSwag | OBQA | PiQA | WinoGrande | Average |
|---|---|---|---|---|---|---|---|---|---|
| | Dense | 50.28 | 16.60 | 55.44 | 62.95 | 29.17 | 43.52 | 19.03 | 39.57 |
| 125M | SparseGPT | 52.33 | 11.60 | 38.17 | 56.15 | 26.61 | 31.52 | 18.60 | 33.57 |
| | Wanda | 50.91 | 11.40 | 37.80 | **56.80** | 26.70 | **31.73** | **18.69** | 33.43 |
| | Thanos ($\alpha = 0$) | 51.54 | 12.40 | 39.76 | 56.64 | 26.66 | **31.73** | 17.15 | 33.70 |
| | **Thanos** ($\alpha = 0.1$) | **52.80** | **14.00** | **43.03** | 56.75 | **26.79** | 31.44 | 17.75 | **34.65** |
| | Dense | 52.64 | 17.60 | 57.61 | 64.53 | 32.03 | 44.11 | 20.82 | 41.33 |
| 350M | SparseGPT | 50.04 | 12.80 | 38.20 | 56.15 | 26.81 | 29.46 | 18.17 | 33.09 |
| | Wanda | 49.72 | 11.60 | 39.17 | **55.88** | 26.43 | **29.67** | 17.66 | 32.88 |
| | Thanos ($\alpha = 0$) | 50.51 | **12.80** | 39.11 | 55.39 | 26.61 | 27.44 | 19.11 | 33.00 |
| | **Thanos** ($\alpha = 0.1$) | **50.59** | 11.40 | **40.03** | 55.44 | **26.94** | 28.79 | **20.14** | **33.33** |

Table 9: Zero-Shot evaluation with different methods for LLaMA-2. Structured 30% sparsity.

| Params | Method | ArcC | ArcE | BoolQ | HellaSwag | OBQA | PiQA | WinoGrande | Average |
|---|---|---|---|---|---|---|---|---|---|
| | Dense | 60.14 | 25.20 | 61.22 | 74.54 | 46.57 | 61.24 | 29.86 | 51.25 |
| 1.1B | SparseGPT | 49.64 | 12.40 | **62.17** | 56.09 | 28.51 | 30.56 | 18.69 | 36.87 |
| | Wanda | 47.04 | 11.60 | 58.93 | 55.01 | 27.19 | 28.54 | **19.45** | 35.39 |
| | Thanos ($\alpha = 0$) | 49.64 | 12.00 | 61.13 | 56.15 | 28.45 | 31.61 | 19.20 | 36.88 |
| | **Thanos** ($\alpha = 0.1$) | **50.51** | **14.00** | 62.11 | **57.34** | 28.97 | **32.28** | 17.75 | **37.57** |
| | Dense | 69.06 | 31.40 | 77.68 | 78.07 | 57.15 | 76.35 | 43.34 | 61.86 |
| 7B | SparseGPT | 52.33 | 14.60 | 54.43 | 56.96 | 29.12 | 32.70 | 20.22 | 37.20 |
| | Wanda | 49.57 | 12.40 | **60.28** | 55.93 | 27.96 | 30.93 | 18.00 | 36.44 |
| | Thanos ($\alpha = 0$) | 49.57 | 12.40 | **60.28** | 55.93 | 27.96 | 30.93 | 18.00 | 36.44 |
| | **Thanos** ($\alpha = 0.1$) | **55.41** | **17.20** | 50.43 | **60.66** | **31.41** | **39.35** | **21.16** | **39.37** |
| | Dense | 71.98 | 35.20 | 80.55 | 79.00 | 60.05 | 79.34 | 48.46 | 64.94 |
| 13B | SparseGPT | 51.38 | 13.80 | 62.08 | 58.43 | 28.88 | 37.42 | 20.31 | 38.90 |
| | Wanda | 51.62 | 12.20 | 62.20 | 55.71 | 27.13 | 29.21 | 19.71 | 36.83 |
| | Thanos ($\alpha = 0$) | 55.56 | 18.60 | 62.20 | 62.35 | 31.33 | 43.18 | **22.10** | 42.19 |
| | **Thanos** ($\alpha = 0.1$) | **58.41** | **19.40** | **63.88** | **63.11** | **33.08** | **45.12** | 21.25 | **43.46** |
| | Dense | 77.90 | 37.20 | 84.00 | 82.21 | 55.35 | 82.95 | 54.44 | 67.72 |
| 70B | Magnitude | 50.51 | 15.60 | 39.20 | 52.23 | 26.95 | 26.00 | 22.35 | 33.26 |
| | Wanda | 56.67 | 16.20 | 60.10 | 62.73 | 33.80 | 46.80 | 19.45 | 42.25 |
| | SparseGPT | 67.32 | 24.80 | 67.35 | 69.64 | 40.90 | 63.15 | 30.80 | 52.00 |
| | Thanos ($\alpha = 0$) | 69.53 | 23.80 | 66.60 | 71.38 | 41.70 | 65.40 | 32.59 | 53.00 |
| | **Thanos** ($\alpha = 0.1$) | **74.51** | **27.40** | **69.75** | **74.59** | **46.35** | **71.50** | **38.99** | **57.58** |

Table 10: Zero-Shot evaluation with different methods for LLaMA-3. Structured 30% sparsity.

| Params | Method | ArcC | ArcE | BoolQ | HellaSwag | OBQA | PiQA | WinoGrande | Average |
|---|---|---|---|---|---|---|---|---|---|
| | Dense | 60.62 | 26.40 | 64.04 | 74.48 | 47.73 | 65.57 | 31.31 | 52.88 |
| 1B | SparseGPT | 49.17 | 13.80 | 57.40 | 54.90 | 26.46 | 28.62 | **20.22** | 35.80 |
| | Wanda | 49.57 | 12.00 | 44.98 | 53.59 | 26.19 | 27.02 | 19.45 | 33.26 |
| | Thanos ($\alpha = 0$) | 49.88 | 13.80 | 58.84 | **56.42** | 27.75 | 32.24 | **20.22** | 37.02 |
| | **Thanos ($\alpha = 0.1$)** | **51.54** | **16.40** | **59.42** | 55.88 | **27.89** | **32.58** | 19.62 | **37.62** |
| | Dense | 69.93 | 31.20 | 73.21 | 76.66 | 55.26 | 74.58 | 42.32 | 60.45 |
| 3B | SparseGPT | 50.67 | 14.00 | **59.79** | 56.91 | 28.51 | 33.04 | 18.77 | 37.38 |
| | Wanda | 50.83 | 13.40 | 58.72 | 54.30 | 26.96 | 29.25 | 19.28 | 36.10 |
| | **Thanos ($\alpha = 0$)** | 52.41 | 13.20 | 59.76 | 57.24 | 29.90 | **35.44** | 18.52 | **38.07** |
| | Thanos ($\alpha = 0.1$) | **52.80** | **14.40** | 52.78 | **58.38** | **30.13** | 34.34 | **20.65** | 37.64 |
| | Dense | 72.77 | 34.80 | 81.31 | 79.71 | 60.19 | 80.13 | 50.51 | 65.63 |
| 8B | SparseGPT | 51.93 | 14.60 | 62.51 | 57.67 | 29.88 | 35.94 | 20.22 | 38.97 |
| | Wanda | 51.30 | 12.80 | 62.11 | 57.34 | 27.64 | 32.87 | 17.32 | 37.34 |
| | Thanos ($\alpha = 0$) | 53.43 | 16.20 | **63.73** | 59.14 | 31.38 | 40.82 | 20.31 | 40.72 |
| | **Thanos ($\alpha = 0.1$)** | **57.38** | **16.60** | 59.24 | **61.70** | **32.66** | **42.34** | **20.39** | **41.47** |
| | Dense | 80.43 | 38.20 | 85.23 | 82.43 | 66.35 | 86.91 | 60.24 | 71.40 |
| 70B | Magnitude | 49.88 | 16.20 | 59.17 | 53.92 | 25.71 | 26.22 | 21.33 | 36.06 |
| | Wanda | 50.91 | 17.00 | 57.71 | 66.70 | 33.77 | 51.39 | 24.15 | 43.09 |
| | SparseGPT | 56.27 | 17.60 | 61.22 | 69.59 | 37.33 | 58.80 | 29.01 | 47.12 |
| | Thanos ($\alpha = 0$) | 70.96 | 22.40 | **68.17** | 71.44 | 39.40 | 61.66 | 29.78 | 51.97 |
| | **Thanos ($\alpha = 0.1$)** | **73.56** | **26.00** | 67.52 | **74.16** | **44.07** | **66.29** | **35.24** | **55.26** |

### E.3 SEMI-STRUCTURED 4:8 SPARSITY

Table 11: Zero-Shot evaluation with different methods for OPT. Semi-structured 4:8 sparsity.

| Params | Method | ArcC | ArcE | BoolQ | HellaSwag | OBQA | PiQA | WinoGrande | Average |
|---|---|---|---|---|---|---|---|---|---|
| | Dense | 50.28 | 16.60 | 55.44 | 62.95 | 29.17 | 43.52 | 19.03 | 39.57 |
| 125M | Magnitude | 51.14 | 13.40 | 51.22 | 58.32 | 27.53 | 34.72 | 18.09 | 36.35 |
| | **SparseGPT** | 50.59 | **15.80** | 61.87 | 60.77 | **27.94** | 38.47 | 19.03 | **39.21** |
| | Wanda | **52.01** | 12.60 | **61.96** | 60.07 | 27.82 | 37.33 | **19.54** | 38.76 |
| | Thanos ($\alpha = 0$) | 50.43 | 13.00 | 61.87 | **61.32** | 27.86 | **39.27** | 18.86 | 38.94 |
| | Thanos ($\alpha = 0.1$) | 51.62 | 14.20 | 60.52 | 61.04 | 27.82 | 39.23 | 18.09 | 38.93 |
| | Dense | 52.64 | 17.60 | 57.61 | 64.53 | 32.03 | 44.11 | 20.82 | 41.33 |
| 350M | Magnitude | 50.99 | 11.20 | 37.92 | 58.38 | 27.73 | 33.08 | 16.72 | 33.72 |
| | **SparseGPT** | 51.54 | **14.40** | **57.52** | 60.94 | 29.58 | 36.66 | 19.71 | **38.62** |
| | Wanda | 50.99 | 11.20 | 53.73 | 60.28 | 28.69 | 36.07 | 17.92 | 36.98 |
| | Thanos ($\alpha = 0$) | 49.64 | 11.60 | 52.32 | 58.71 | 28.52 | 32.45 | **19.45** | 36.10 |
| | Thanos ($\alpha = 0.1$) | **52.33** | 13.00 | 56.09 | **60.99** | 29.62 | 38.17 | 18.86 | 38.44 |

Table 12: Zero-Shot evaluation with different methods for LLaMA-2. Semi-structured 4:8 sparsity.

| Params | Method | ArcC | ArcE | BoolQ | HellaSwag | OBQA | PiQA | WinoGrande | Average |
|---|---|---|---|---|---|---|---|---|---|
| | Dense | 60.14 | 25.20 | 61.22 | 74.54 | 46.57 | 61.24 | 29.86 | 51.25 |
| 1.1B | Magnitude | 54.30 | 17.40 | 58.50 | 65.61 | 35.38 | 46.21 | 22.01 | 42.77 |
| | SparseGPT | 56.27 | 19.40 | 62.35 | 67.79 | 36.52 | 49.75 | 21.84 | 44.85 |
| | Wanda | 55.17 | 17.40 | 62.35 | 66.54 | 34.97 | 46.25 | 22.35 | 43.58 |
| | **Thanos** ($\alpha = 0$) | **58.09** | **20.40** | 62.26 | **68.34** | 36.75 | 48.74 | **23.46** | **45.43** |
| | Thanos ($\alpha = 0.1$) | 57.93 | 19.20 | **62.78** | 68.01 | **37.21** | **49.92** | 22.44 | 45.36 |
| | Dense | 69.06 | 31.40 | 77.68 | 78.07 | 57.15 | 76.35 | 43.34 | 61.86 |
| 7B | Magnitude | 62.27 | 26.00 | 63.03 | 72.20 | 50.05 | 64.81 | 36.01 | 53.48 |
| | SparseGPT | **67.25** | **28.20** | 69.69 | 74.70 | 48.29 | 68.69 | 34.81 | 55.95 |
| | Wanda | 66.77 | 26.40 | 72.42 | 74.54 | 47.04 | 67.00 | 34.56 | 55.53 |
| | Thanos ($\alpha = 0$) | 66.61 | **28.20** | 71.25 | 74.37 | 48.11 | **68.98** | **36.09** | 56.23 |
| | **Thanos** ($\alpha = 0.1$) | 66.38 | **28.20** | **76.15** | **75.35** | **49.20** | 68.69 | 35.84 | **57.11** |
| | Dense | 71.98 | 35.20 | 80.55 | 79.00 | 60.05 | 79.34 | 48.46 | 64.94 |
| 13B | Magnitude | 64.40 | 26.00 | 63.33 | 74.10 | **53.96** | 68.43 | 35.84 | 55.15 |
| | SparseGPT | 69.69 | 28.40 | 80.70 | 75.68 | 52.25 | **74.07** | **41.98** | 60.40 |
| | Wanda | 69.22 | 30.00 | 79.91 | 75.95 | 52.23 | 73.78 | 39.85 | 60.13 |
| | Thanos ($\alpha = 0$) | **70.56** | 30.60 | 81.56 | 75.73 | 52.25 | 73.95 | 41.21 | 60.84 |
| | **Thanos** ($\alpha = 0.1$) | 70.01 | **31.00** | **81.62** | **76.22** | 53.57 | 73.70 | 41.21 | **61.05** |
| | Dense | 77.90 | 37.20 | 84.00 | 82.21 | 55.35 | 82.95 | 54.44 | 67.72 |
| 70B | Magnitude | 74.74 | 34.40 | 70.70 | 78.67 | 51.75 | 76.05 | 46.59 | 61.84 |
| | Wanda | 75.69 | 35.60 | 84.00 | 80.47 | 52.50 | 80.20 | 50.17 | 65.52 |
| | SparseGPT | 76.95 | 35.20 | 83.90 | 80.20 | 52.25 | 79.90 | 51.02 | 65.63 |
| | Thanos ($\alpha = 0$) | **77.51** | 35.00 | 83.65 | 80.52 | 51.95 | 80.45 | 50.60 | 65.67 |
| | **Thanos** ($\alpha = 0.1$) | 77.27 | **35.80** | **85.45** | **81.34** | **53.40** | **81.20** | **51.62** | **66.58** |

Table 13: Zero-Shot evaluation with different methods for LLaMA-3. Semi-structured 4:8 sparsity.

| Params | Method | ArcC | ArcE | BoolQ | HellaSwag | OBQA | PiQA | WinoGrande | Average |
|--------|--------|------|------|-------|-----------|------|------|------------|---------|
| | Dense | 60.62 | 26.40 | 64.04 | 74.48 | 47.73 | 65.57 | 31.31 | 52.88 |
| 1B | Magnitude | 50.36 | 11.60 | 38.32 | 55.11 | 26.79 | 30.30 | 19.03 | 33.07 |
| | SparseGPT | **57.70** | 19.00 | 62.05 | 65.72 | 34.82 | 49.28 | 22.18 | 44.39 |
| | Wanda | 51.38 | 14.40 | 61.90 | 62.51 | 30.49 | 42.34 | 19.71 | 40.39 |
| | Thanos ($\alpha = 0$) | 56.43 | 19.40 | 62.57 | 65.72 | 34.68 | 50.38 | 22.87 | 44.58 |
| | **Thanos ($\alpha = 0.1$)** | 54.62 | **19.60** | **63.00** | **66.00** | **35.98** | **51.89** | **23.12** | **44.89** |
| | Dense | 69.93 | 31.20 | 73.21 | 76.66 | 55.26 | 74.58 | 42.32 | 60.45 |
| 3B | Magnitude | 54.38 | 15.80 | 54.34 | 64.09 | 31.77 | 45.24 | 22.10 | 41.10 |
| | SparseGPT | 63.93 | 21.40 | **70.28** | 70.29 | 42.87 | 60.48 | 29.18 | 51.20 |
| | Wanda | 58.88 | 20.20 | 65.50 | 68.28 | 38.53 | 59.01 | 28.16 | 48.37 |
| | Thanos ($\alpha = 0$) | 63.77 | 23.00 | 69.88 | 70.08 | 42.58 | 61.24 | 30.20 | 51.54 |
| | **Thanos ($\alpha = 0.1$)** | **64.40** | **23.40** | 69.45 | **70.84** | **43.44** | **63.17** | **31.23** | **52.28** |
| | Dense | 72.77 | 34.80 | 81.31 | 79.71 | 60.19 | 80.13 | 50.51 | 65.63 |
| 8B | Magnitude | 52.57 | 19.20 | 44.43 | 65.29 | 37.13 | 53.24 | 26.54 | 42.63 |
| | SparseGPT | 69.93 | 24.40 | 74.34 | 73.72 | 47.91 | 68.22 | 34.56 | 56.15 |
| | Wanda | 66.61 | 24.40 | 71.38 | 71.55 | 44.07 | 65.87 | 34.39 | 54.04 |
| | Thanos ($\alpha = 0$) | 69.06 | 25.60 | **78.07** | 73.50 | 47.75 | 68.81 | 35.41 | 56.89 |
| | **Thanos ($\alpha = 0.1$)** | **71.03** | **27.00** | 75.69 | **74.76** | **48.83** | **69.99** | **37.71** | **57.86** |
| | Dense | 80.43 | 38.20 | 85.23 | 82.43 | 66.35 | 86.91 | 60.24 | 71.40 |
| 70B | Magnitude | 67.17 | 27.80 | 74.50 | 72.52 | 46.44 | 71.13 | 38.91 | 56.92 |
| | Wanda | 73.88 | 34.20 | 81.16 | 80.36 | 59.81 | 81.40 | 51.02 | 65.98 |
| | SparseGPT | 79.32 | 35.00 | **85.50** | 80.03 | 59.54 | 81.99 | 52.30 | 67.67 |
| | Thanos ($\alpha = 0$) | 78.93 | 34.40 | 85.20 | 80.36 | 60.02 | 82.20 | 52.30 | 67.63 |
| | **Thanos ($\alpha = 0.1$)** | **80.43** | **36.00** | 84.95 | **81.07** | **61.32** | **82.45** | **55.12** | **68.76** |

## E.4 SEMI-STRUCTURED 2:4 SPARSITY

Table 14: Zero-Shot evaluation with different methods for OPT. Semi-structured 2:4 sparsity.

| Params | Method | ArcC | ArcE | BoolQ | HellaSwag | OBQA | PiQA | WinoGrande | Average |
|--------|--------|------|------|-------|-----------|------|------|------------|---------|
| | Dense | 50.28 | 16.60 | 55.44 | 62.95 | 29.17 | 43.52 | 19.03 | 39.57 |
| 125M | Magnitude | 50.67 | 13.40 | 61.04 | 57.73 | 27.14 | 32.28 | 17.83 | 37.16 |
| | SparseGPT | **51.85** | 14.00 | **62.23** | 58.54 | 27.25 | 36.66 | **19.54** | 38.58 |
| | Wanda | 49.72 | 14.20 | 61.99 | 59.25 | 27.60 | 36.24 | 18.60 | 38.23 |
| | Thanos ($\alpha = 0$) | 51.54 | 14.20 | 61.77 | 59.47 | **27.71** | 38.05 | 18.09 | 38.69 |
| | **Thanos ($\alpha = 0.1$)** | 51.62 | **15.40** | 61.28 | **59.68** | 27.67 | **38.80** | 18.09 | **38.94** |
| | Dense | 52.64 | 17.60 | 57.61 | 64.53 | 32.03 | 44.11 | 20.82 | 41.33 |
| 350M | Magnitude | 50.91 | 11.60 | 59.17 | 57.29 | 27.12 | 31.52 | 16.81 | 36.35 |
| | **SparseGPT** | 49.96 | 11.20 | **59.51** | 59.47 | 28.74 | 34.34 | 18.26 | **37.35** |
| | Wanda | 50.12 | 12.60 | 48.47 | 58.60 | 28.10 | 34.72 | 16.81 | 35.63 |
| | Thanos ($\alpha = 0$) | 49.80 | **15.00** | 47.25 | 57.73 | 28.01 | 32.24 | 19.54 | 35.65 |
| | Thanos ($\alpha = 0.1$) | **51.93** | 11.80 | 47.61 | **60.01** | **29.17** | **37.08** | **19.97** | 36.80 |

Table 15: Zero-Shot evaluation with different methods for LLaMA-2. Semi-structured 2:4 sparsity.

| Params | Method | ArcC | ArcE | BoolQ | HellaSwag | OBQA | PiQA | WinoGrande | Average |
|---|---|---|---|---|---|---|---|---|---|
| | Dense | 60.14 | 25.20 | 61.22 | 74.54 | 46.57 | 61.24 | 29.86 | 51.25 |
| 1.1B | Magnitude | 52.17 | 15.60 | 39.33 | 62.13 | 32.11 | 40.66 | 19.28 | 37.33 |
| | SparseGPT | 55.96 | 17.20 | 61.83 | 64.53 | 33.66 | **46.80** | 21.42 | 43.06 |
| | Wanda | 53.12 | 14.60 | 61.59 | 63.33 | 31.63 | 41.46 | 20.65 | 40.91 |
| | **Thanos** ($\alpha = 0$) | **56.20** | **17.40** | **62.20** | 65.18 | **33.76** | 45.50 | **22.01** | **43.18** |
| | Thanos ($\alpha = 0.1$) | 54.54 | 16.20 | 62.14 | **65.89** | 34.59 | 45.33 | 20.99 | 42.81 |
| | Dense | 69.06 | 31.40 | 77.68 | 78.07 | 57.15 | 76.35 | 43.34 | 61.86 |
| 7B | Magnitude | 60.93 | 21.80 | 59.82 | 70.08 | 45.43 | 61.87 | 30.20 | 50.02 |
| | SparseGPT | 65.51 | 24.60 | 69.51 | 71.27 | 43.52 | 63.43 | 29.95 | 52.54 |
| | Wanda | 62.67 | 24.00 | 68.35 | 70.62 | 41.45 | 61.91 | 30.46 | 51.35 |
| | Thanos ($\alpha = 0$) | **66.14** | **26.20** | 66.09 | 72.14 | 44.16 | 64.60 | 31.23 | 52.94 |
| | **Thanos** ($\alpha = 0.1$) | 65.51 | 25.80 | **74.16** | **72.42** | **45.69** | **65.45** | **32.68** | **54.53** |
| | Dense | 71.98 | 35.20 | 80.55 | 79.00 | 60.05 | 79.34 | 48.46 | 64.94 |
| 13B | Magnitude | 61.96 | 23.20 | 65.66 | 71.71 | 50.11 | 62.29 | 31.74 | 52.38 |
| | SparseGPT | 69.93 | 27.40 | **79.66** | 73.99 | 47.85 | 69.65 | 36.52 | 57.86 |
| | Wanda | 67.56 | 25.40 | 76.06 | 74.16 | 46.42 | 68.27 | 34.90 | 56.11 |
| | Thanos ($\alpha = 0$) | 69.77 | **27.80** | 76.48 | 74.05 | 47.85 | 69.40 | 36.52 | 57.41 |
| | **Thanos** ($\alpha = 0.1$) | **71.19** | **27.80** | 79.57 | **74.65** | **49.84** | **70.71** | **38.74** | **58.93** |
| | Dense | 77.90 | 37.20 | 84.00 | 82.21 | 55.35 | 82.95 | 54.44 | 67.72 |
| 70B | Magnitude | 74.66 | 35.20 | 73.60 | 77.69 | 50.00 | 75.95 | 45.39 | 61.79 |
| | Wanda | 74.59 | 34.80 | 81.50 | 79.11 | 51.05 | 79.45 | 47.27 | 63.97 |
| | SparseGPT | 76.16 | 32.40 | 81.35 | 78.89 | 51.35 | 79.10 | 48.12 | 63.91 |
| | Thanos ($\alpha = 0$) | 76.48 | 33.20 | 79.95 | 79.38 | 51.45 | 79.55 | 47.95 | 63.99 |
| | **Thanos** ($\alpha = 0.1$) | **77.19** | **35.40** | **84.00** | **80.20** | **53.40** | **80.05** | **49.32** | **65.65** |

Table 16: Zero-Shot evaluation with different methods for LLaMA-3. Semi-structured 2:4 sparsity.

| Params | Method | ArcC | ArcE | BoolQ | HellaSwag | OBQA | PiQA | WinoGrande | Average |
|--------|--------|------|------|-------|-----------|------|------|------------|---------|
| | Dense | 60.62 | 26.40 | 64.04 | 74.48 | 47.73 | 65.57 | 31.31 | 52.88 |
| 1B | Magnitude | 51.85 | 11.80 | 38.53 | 54.19 | 26.15 | 27.90 | 19.37 | 32.83 |
| | **SparseGPT** | **55.49** | 16.40 | 62.11 | 62.13 | 32.58 | 46.63 | **23.04** | **42.63** |
| | Wanda | 51.62 | 14.00 | 56.36 | 58.60 | 28.26 | 36.28 | 18.26 | 37.63 |
| | Thanos ($\alpha = 0$) | 53.83 | 16.40 | **62.14** | 62.89 | 32.60 | 45.71 | 20.82 | 42.06 |
| | Thanos ($\alpha = 0.1$) | 53.35 | **16.80** | 61.96 | **63.38** | **33.25** | **47.56** | 21.33 | 42.52 |
| | Dense | 69.93 | 31.20 | 73.21 | 76.66 | 55.26 | 74.58 | 42.32 | 60.45 |
| 3B | Magnitude | 50.43 | 14.40 | 38.47 | 60.55 | 28.39 | 37.54 | 19.37 | 35.59 |
| | SparseGPT | 59.04 | 21.80 | 67.09 | 68.61 | 38.50 | 57.41 | 27.39 | 48.55 |
| | Wanda | 55.96 | 16.80 | 63.61 | 65.18 | 34.04 | 50.21 | 25.43 | 44.46 |
| | Thanos ($\alpha = 0$) | 60.54 | 21.20 | 67.86 | 68.12 | 38.45 | **57.70** | 25.77 | 48.52 |
| | **Thanos** ($\alpha = 0.1$) | **62.43** | **23.40** | **69.72** | **68.77** | **39.65** | 56.78 | **28.41** | **49.88** |
| | Dense | 72.77 | 34.80 | 81.31 | 79.71 | 60.19 | 80.13 | 50.51 | 65.63 |
| 8B | Magnitude | 53.20 | 15.40 | 38.32 | 61.64 | 30.78 | 40.45 | 21.50 | 37.33 |
| | SparseGPT | 65.67 | 22.20 | 73.58 | 70.08 | 43.21 | 62.63 | 31.91 | 52.75 |
| | Wanda | 59.98 | 18.60 | 66.15 | 67.19 | 37.51 | 56.02 | 26.37 | 47.40 |
| | Thanos ($\alpha = 0$) | 65.75 | 21.00 | 75.54 | 70.62 | 42.92 | 62.84 | 30.29 | 52.71 |
| | **Thanos** ($\alpha = 0.1$) | **66.30** | **23.20** | **75.63** | **72.14** | **44.81** | **65.32** | **33.36** | **54.39** |
| | Dense | 80.43 | 38.20 | 85.23 | 82.43 | 66.35 | 86.91 | 60.24 | 71.40 |
| 70B | Magnitude | 58.80 | 24.20 | 65.78 | 72.42 | 47.03 | 69.32 | 37.80 | 53.62 |
| | Wanda | 71.59 | 29.80 | 77.65 | 79.33 | 55.90 | 78.32 | 47.27 | 62.84 |
| | SparseGPT | 77.35 | 30.80 | 83.79 | 78.62 | 55.07 | 79.00 | 47.61 | 64.61 |
| | Thanos ($\alpha = 0$) | 77.11 | 31.40 | 82.54 | 79.22 | 55.45 | 77.53 | 47.35 | 64.37 |
| | **Thanos** ($\alpha = 0.1$) | **79.01** | **32.40** | **84.40** | **80.09** | **58.50** | **79.46** | **50.68** | **66.36** |

# F    FORMULATION OF THE PROBLEM

## F.1    GENERAL PRUNING ALGORITHM

In this work we will consider the following generic pruning Algorithm 3, which is used in many data-aware pruning papers (Frantar & Alistarh, 2023; Sun et al., 2023):

---
**Algorithm 3** General Pruning Algorithm for LLM
---
1: **Initialization:** Input tensor, linear layers, sparsity ratio.
2: **for** every **block** of the model **do**
3:     Compute inputs of linear layers by a forward pass of the **block**.
4:     **for** every linear layer inside the **block do**
5:         Prune layer $W$ using **some algorithm**, to a fixed sparsity ratio.
6:     **end for**
7:     Compute the input for the next block by a forward pass of the pruned **block**.
8: **end for**
---

The pruning is performed sequentially, block by block. In the beginning we consider the first block, for witch we make a forward pass to compute the input matrix $X$ for every linear layer $W$ inside the block, then we perform pruning of every linear layer inside the block to the desired sparsity ratio by using **some algorithm**. So, every linear layer $W$ is pruned independently of each other by the same procedure to the same level of sparsity.

As soon as we pruned all linear layers of the first block, we make a second forward pass through the pruned block to compute the modified input for the second block. Then we repeat the same steps, mentioned before, but for the next block of LLM and so on. As a result we have a fully pruned model.

The most interesting part of the aforementioned algorithm is the choice of the pruning algorithm.

## F.2    OBJECTIVE FUNCTION

Post-training compression typically involves breaking down the task of compressing the entire model into smaller, layer-wise sub-problems. Recent studies in this field have shown that pruning with limited data-based objective can be very efficient for pruning of LLMs. In this objective we aim to minimize the difference between the output of linear layer before and after pruning (Hubara et al., 2021a).

Let us now consider a single layer $W \in \mathbb{R}^{c \times b}$ and the corresponding input matrix $X \in \mathbb{R}^{b \times a}$. We denote $\widehat{W} \in \mathbb{R}^{c \times b}$ as a pruned weight matrix, then the objective function $\widehat{f} : \mathbb{R}^{c \times b} \to \mathbb{R}$ will be:

$$\widehat{f}(\widehat{W}) := \|(\widehat{W} - W)X\|_F^2, \tag{12}$$

where $\| \cdot \|_F$ denotes the Frobenius norm, that is defined for every real matrix $A$ by

$$\|A\|_F := \sqrt{\sum_{i=1} \sum_{j=1} |A_{ij}|^2}. \tag{13}$$

So, the weight matrix $\widehat{W}$ is constructed by removal some weights from $W$ and adjusting all remaining (non-pruned) weights to minimize the objective (12).

In this paper we work with the objective function (12) and suggest a novel method of minimizing it. The main problem of this objective is that the pruned matrix $\widehat{W}$ is forced to have a fixed sparsity ratio, so the optimization problem becomes constrained.

Note, that the problem of minimizing the objective (12) is discrete because we need to minimize it over a discrete set of pruning masks – matrices with zeros and ones, that indicates what parameters should be removed. This makes the problem hard to solve, that is why we will split it into two separate problem: find the best pruning mask and then find the best adjustment of remaining parameters.

### F.3 CONSTRAINED OPTIMIZATION PROBLEM

Let $M \in \{0,1\}^{c \times b}$ be a pruning matrix – matrix of those entries of $W$, that should be pruned. Every element $M_{ij} \in M$ can be either zero or one: $M_{ij} \in \{0,1\}$: $M_{ij} = 1$ means we will zero out the weight $W_{ij}$.

Let us denote $p$ as a desired sparsity ratio after the pruning by the Algorithm 3). We define the sparsity ratio $p \in [0,1)$ in the following way:

$$p := \frac{\|M\|_F^2}{cb}, \tag{14}$$

where $\|M\|_F^2 \in \{0, \cdots, cb\}$ indicates how much parameters should be removed from $W \in \mathbb{R}^{c \times b}$.

Note that

- $p = 0$ means that a matrix $M$ is sparse (zero matrix) since $p = 0 = \frac{\|M\|_F^2}{cb}$, hence $\|M\|_F^2 = 0$ – all elements of $M$ are zero.

- $p = 1$ means that a matrix $M$ is dense (all elements of $M$ are equal to one) since $p = 1 = \frac{\|M\|_F^2}{cb}$, hence $\|M\|_F^2 = cb$, therefore there are no parameters of $M$ that are equal to zero.

- $p = 1/2$ means that half of the elements from $M$ are zero. Indeed, $p = \frac{1}{2} = \frac{\|M\|_F^2}{cb}$, therefore $\|M\|_F^2 = \frac{1}{2}cb$, so half of the elements are equal to zero.

- $\widehat{W}$ is strict to have $p\%$ sparsity. A denser $M$ results in a more sparse $\widehat{W}$ because the more parameters we prune, the less non-zero parameters we have in the final matrix $\widehat{W}$.

In essence, $p$ indicates how strong we want to prune the linear layer $W$. The larger $p$ is, the more information we lose.

We can write $\widehat{W}$ in the following form:

$$\widehat{W} = (1 - M) \odot \widetilde{W}, \tag{15}$$

where $\widetilde{W} \in \mathbb{R}^{c \times b}$ and $(\odot)$ denotes an element-wise product (Hadamard product). Such a decomposition of $\widehat{W}$ is useful because $\widetilde{W}$ does not have to have a given level of sparsity.

By using decomposition (15) of $\widehat{W}$, we can rewrite the function $\widehat{f}(\widehat{W})$ by the following way:

$$\widehat{f}(\widehat{W}) \overset{(12)}{:=} \|(\widehat{W} - W)X\|_F^2 \overset{(15)}{=} \left\| \left( (1 - M) \odot \widetilde{W} - W \right) X \right\|_F^2 =: \widetilde{f}(\widetilde{W}, M). \tag{16}$$

With the aforementioned notations we introduce the following discrete constrained optimization problem of pruning a linear layer $W$:

$$\min_{\widetilde{W} \in \mathbb{R}^{c \times b}, \, M \in \{0,1\}^{c \times b}} \left[ \widetilde{f}(\widetilde{W}, M) \overset{(16)}{:=} \left\| \left( (1 - M) \odot \widetilde{W} - W \right) X \right\|_F^2 \right], \tag{17}$$

$$\text{subject to} \quad \|M\|_F^2 \overset{(14)}{=} \lfloor pcb \rfloor,$$

where $\lfloor \cdot \rfloor : \mathbb{R} \to \mathbb{Z}$ is a floor function, defined by

$$\lfloor x \rfloor := \max \{ n \in \mathbb{Z} \mid n \leq x \}.$$

We use a floor function in constraints of (17) because $\|M\|_F^2 \in \{0, \cdots, cb\}$ can only take integer values.

The problem (17) is complicated because of the discrete nature of its sparsity constraints. This makes this problem NP-hard (Blumensath & Davies, 2008). Consequently, finding an exact solution for larger layers becomes computationally infeasible, prompting most existing methods to rely on approximation techniques.

A widely adopted strategy involves decomposing the compression task into two steps: selecting a pruning mask and reconstructing the remaining weights (He et al., 2018; Kwon et al., 2022; Hubara

et al., 2021a). In this approach, a pruning mask $M$ is first chosen based on some metric, such as weight magnitude (Zhu & Gupta, 2017). After the mask is determined, the focus shifts to optimizing the unpruned weights while keeping the mask fixed. Notably, with a fixed mask, the problem reduces to a linear least-squares optimization, which can be solved efficiently.

### F.4 PRUNING OF A SINGLE PARAMETER OF THE LAYER

We will make the following simplification for the problem (17) and its constraints to be able to effectively solve the optimal pruning problem: let us solve this problem for only removing a single weight from $W$.

That is being said: our task is to solve the problem (17) with only one $M_{kq} = 1$ ($\widehat{W}_{kq} = 0$) for some $k \in \{1, \cdots, c\}$ and $q \in \{1, \cdots, b\}$. For all $i \neq k$ and $j \neq q$, $M_{ij} = 0$.

In a practical scenario, we need to prune the layer $W$ to achieve a target sparsity level of $p\%$. For the problem defined in Equation (17), where multiple weights must be pruned, an effective approach is to apply the single-weight pruning solution in a greedy, iterative manner. This approach incrementally removes the least important weights, one by one, until the desired sparsity level is reached.

The loss function can be written as

$$\widehat{f}(\widehat{W}) \stackrel{(12)}{=} \|(\widehat{W} - W)X\|_F^2 = \|((W + \Delta) - W)X\|_F^2 = \|\Delta X\|_F^2, \tag{18}$$

where $\Delta := \widehat{W} - W$, $\Delta \in \mathbb{R}^{c \times b}$ is a change of the weight matrix $W$:

$$\Delta := \begin{pmatrix} \Delta_{11} & \cdots & \Delta_{1b} \\ \vdots & \ddots & \vdots \\ \Delta_{c1} & \cdots & \Delta_{cb} \end{pmatrix} = \begin{pmatrix} \Delta_{1:} \\ \vdots \\ \Delta_{c:} \end{pmatrix},$$

where $\Delta_{i:} \in \mathbb{R}^{1 \times b}$ is the $i^{\text{th}}$ row of the matrix $\Delta$.

The objective (18) can be rewritten in the following form:

$$\widehat{f}(\widehat{W}) \stackrel{(12)}{=} \|(\widehat{W} - W)X\|_F^2 \stackrel{(18)}{=} \|\Delta X\|_F^2 \stackrel{(13)}{=} \sum_{i=1}^{c} \|\Delta_{i:}X\|_2^2 = \sum_{i=1}^{c} g(\Delta_{i:}) =: f(\Delta), \tag{19}$$

where $\|\cdot\|_2$ denotes $l^2$-norm, that is defined for every vector $x$:

$$\|x\|_2 := \sqrt{\sum_{i=1}^{} |x_i|^2},$$

and the function $g : \mathbb{R}^{1 \times b} \to \mathbb{R}$ is defined by

$$g(\delta) := \|\delta X\|_2^2. \tag{20}$$

We can see that $g(\Delta_{i:})$ depends only on the $i^{\text{th}}$ row of the matrix $\Delta$.

As mentioned before, our goal is to optimally prune a single weight from the layer $W \in \mathbb{R}^{c \times b}$. Let us denote this weight as $W_{kq}$, so we aim to delete the weight from the $k^{\text{th}}$ row and $q^{\text{th}}$ column. Hence $\Delta_{kq} + W_{kq} = 0$ or, in another form, $\Delta_{k:}e^q + W_{kq} = 0$, where $e^q \in \mathbb{R}^{b \times 1}$ is a unit column-vector with 1 at the $q^{\text{th}}$ position.

Note that neither $k \in \{1, \cdots, c\}$ nor $q \in \{1, \cdots, b\}$ are known in advance – they should be found based on some criteria of optimality of pruning.

Then we aim to solve the following optimization problem

$$\min_{\Delta \in \mathbb{R}^{c \times b}} \left[ f(\Delta) \stackrel{(19)}{:=} \sum_{i=1}^{c} g(\Delta_{i:}) \right], \tag{21}$$

$$\text{subject to} \quad \Delta_{k:}e^q + W_{kq} = 0.$$

We can simplify this problem by using the following the Lemma 1:

**Lemma 1.** *The optimal solution to the problem (21) satisfy:*

$$\Delta_{i:} = 0 \in \mathbb{R}^{1 \times b} \quad \text{for all } i \neq k. \tag{22}$$

*Proof.* As we mentioned before, we aim to remove a single weight from some row $k \in \{1, \cdots, c\}$ and some column $q \in \{1, \cdots, b\}$.

Since all entries of the sum in (19) are larger than zero, the solution for the problem (21) will be reached when each $g(\Delta_{i:})$ takes its minimal value. Now let us consider all $i \neq k$. The function $g(\Delta_{i:})$ reaches its minimal value with $\Delta_{i:} = 0 \in \mathbb{R}^{1 \times b}$. Moreover $\Delta_{i:} = 0 \in \mathbb{R}^{1 \times b}$ satisfies the constraints of (21) for all $i \neq k$.

Hence, the optimal solution to the problem (21) satisfies $\Delta_{i:} = 0 \in \mathbb{R}^{1 \times b}$ for all $i \neq k$. $\qquad\square$

By the Lemma 1 we have:

$$f(\Delta) \overset{(19)}{:=} \sum_{i=1}^{c} g(\Delta_{i:}) = g(\Delta_{k:}) + \sum_{i \neq k} g(\Delta_{i:}) \overset{\text{(Lemma 1)}}{=} g(\Delta_{k:}) + \sum_{i \neq k} g(0) = g(\Delta_{k:}). \tag{23}$$

For simplicity let us denote $\Delta_{k:} =: \delta$, so $\delta \in \mathbb{R}^{1 \times b}$ is a row-vector. In this notation $g(\Delta_{k:}) = \|\Delta_{k:}X\|_2^2 = \|\delta X\|_2^2 = g(\delta)$. In addition, let us denote the $k^{\text{th}}$ row of $W$ as $w := W_{k:}$, $w \in \mathbb{R}^{1 \times b}$ is a row-vector. With the aforementioned notation,

$$f(\Delta) \overset{(23)}{=} g(\Delta_{k:}) = g(\delta).$$

Then the problem (21), can be written in the following form:

$$\min_{\delta \in \mathbb{R}^{1 \times b}} \left[ g(\delta) \overset{(20)}{:=} \|\delta X\|_2^2 \right], \tag{24}$$
$$\text{subject to} \quad \delta e^q + w_q = 0.$$

### F.5   Objective function for $d$ calibration samples

In real applications we need to consider more than one calibration sample to obtain reasonable perplexity. That is why the input tensor $\mathcal{X} \in \mathbb{R}^{d \times b \times a}$ will have additional dimension of size $d$ – the number of calibration samples.

Let us split the tensor $\mathcal{X}$ into $d$ matrices of size $b \times a$: $\mathcal{X} = \{X^1, \cdots, X^d\}$, where $X^l \in \mathbb{R}^{b \times a}$ for $n \in \{1, \cdots, d\}$, then the loss function $F : \mathbb{R}^{c \times b} \to \mathbb{R}$ will be the average between loss functions which were computed for each $X^l$:

$$F(\Delta) := \frac{1}{d} \sum_{l=1}^{d} \|\Delta X^l\|_F^2 \overset{(13)}{=} \frac{1}{d} \sum_{l=1}^{d} \sum_{i=1}^{c} \|\Delta_{i:} X^l\|_2^2 =$$

$$= \frac{1}{d} \sum_{i=1}^{c} \sum_{l=1}^{d} \|\Delta_{i:} X^l\|_2^2 = \sum_{i=1}^{c} G(\delta),$$

where $G : \mathbb{R}^{1 \times b} \to \mathbb{R}$:

$$G(\delta) := \frac{1}{d} \sum_{l=1}^{d} \|\delta X^l\|_2^2. \tag{25}$$

Similarly to how we derive the problem (24) in (sec. F.4), we come to the following formulation:

$$\min_{\delta \in \mathbb{R}^{1 \times b}} \left[ G(\delta) \overset{(25)}{:=} \sum_{l=1}^{d} \|\delta X^l\|_2^2 \right], \tag{26}$$
$$\text{subject to} \quad \delta e^q + w_q = 0.$$

## G  BACKGROUND AND RELATED WORK

### G.1  MAGNITUDE PRUNING

One of the simplest and most intuitive methods for pruning LLMs is Magnitude pruning (Algorithm 4), first introduced by (Han et al., 2015), is a widely used method for introducing sparsity in neural networks. This technique eliminates individual weights by evaluating their magnitudes, discarding those that fall below a specific threshold.

---

**Algorithm 4** Magnitude pruning algorithm

---

1: **Initialization:** Weight matrix $W \in \mathbb{R}^{c \times b}$, sparsity ratio $p \in [0, 1)$.
2: $M \leftarrow$ mask of $p\%$ weights $W_{ij}$ from $W$ with smallest $|W_{ij}|$.
3: $W \leftarrow W - M \odot W$

---

The core principle behind this technique is that weights with smaller magnitudes contribute less to the overall performance of the model compared to those with larger magnitudes. As such, these smaller weights can be safely removed with minimal impact on the model's accuracy.

The complexity of the Algorithm 4 is

$$T_{\text{Magnitude}} = \mathcal{O}\left(cb \log cb\right). \tag{27}$$

A key advantage of magnitude-based pruning lies in its data-free nature. Since it does not require any input matrix $X$, this method eliminates the need for a calibration dataset, making it straightforward to implement. Additionally, it is highly efficient in terms of computational cost, as the pruning decisions rely solely on the analysis of the model's internal weights.

The threshold can be applied either locally within each layer or globally across the entire model. Although straightforward, magnitude pruning has demonstrated the ability to identify highly sparse networks, as highlighted by the work of (Frankle & Carbin, 2018). Today, it is regarded as a robust baseline for network sparsification, as noted by (Blalock et al., 2020).

However, the performance of magnitude pruning degrades significantly at higher sparsity levels, where a large fraction of the weights are removed. This outcome is surprising, given that magnitude pruning has shown great success on smaller networks. Despite LLMs having far more parameters, often by a factor of 100 to 1000, these additional parameters do not make the pruning process any easier. Instead, the sheer scale introduces new complexities that complicate direct pruning efforts.

For such large models like LLM, data-aware pruning methods which leverage small calibration datasets consistently yield superior results (He et al., 2018; Kwon et al., 2022; Hubara et al., 2021a; Frantar & Alistarh, 2023; Sun et al., 2023). For this reason, our study primarily focuses on data-aware techniques, as they strike a better balance between sparsity and performance, even with limited calibration data.

### G.2  OBD AND OBS

A classical approach to solve the problems (24) and (26) was presented in Optimal Brain Damage (OBD) (Hassibi & Stork, 1992) and Optimal Brain Surgeon (OBS) (LeCun et al., 1989) papers. In this section we will derive the main results from these papers.

Let us approximate the function $g(\delta)$ by the Taylor series up to the third order components near the point 0:

$$g(\delta) = g(0) + \frac{\partial g}{\partial \delta}\delta + \frac{1}{2}\delta H \delta^\top + O(|\delta|^3), \tag{28}$$

where $H \in \mathbb{R}^{b \times b}$ is a Hessian matrix: $H = \frac{\partial^2 g}{\partial \delta^2}$.

The Hessian matrix for our problem can be easily found.

For the objective function in (24), we have

$$g(\delta) \overset{(20)}{:=} \|\delta X\|_2^2 = (\delta X)(\delta X)^\top = \delta X X^\top \delta^\top. \tag{29}$$

The Hessian of $g$ can be computed by the following simple derivations:

$$H := \frac{\partial^2 g(\delta)}{\partial \delta^2} \overset{(29)}{=} \frac{\partial^2}{\partial \delta^2} \left( \delta X X^\top \delta^\top \right) = \frac{\partial}{\partial \delta} \left( 2 X X^\top \delta^\top \right) = 2 X X^\top. \tag{30}$$

Similarly, in case of $d$-samples objective (25) we have

$$G(\delta) \overset{(25)}{:=} \frac{1}{d} \sum_{l=1}^{d} \| \delta X^l \|_2^2 = \frac{1}{d} \sum_{l=1}^{d} (\delta X^l)(\delta X^l)^\top =$$

$$= \frac{1}{d} \sum_{l=1}^{d} \delta X^l (X^l)^\top \delta^\top = \frac{1}{d} \delta \left( \sum_{l=1}^{d} X^l (X^l)^\top \right) \delta^\top, \tag{31}$$

hence, the Hessian will be

$$H_G := \frac{\partial^2 G(\delta)}{\partial \delta^2} \overset{(31)}{=} \frac{\partial^2}{\partial \delta^2} \left( \frac{1}{d} \delta \left( \sum_{l=1}^{d} X^l (X^l)^\top \right) \delta^\top \right) =$$

$$= \frac{1}{d} \frac{\partial}{\partial \delta} \left( 2 \left( \sum_{l=1}^{d} X^l (X^l)^\top \right) \delta^\top \right) = 2 \frac{1}{d} \sum_{l=1}^{d} X^l (X^l)^\top.$$

Note that

- The Hessian is the same for all rows of the matrix $W$ since it does not depend on either $w$ or $\delta$.

- As expected, the Hessian is symmetric positive definite and hence its inverse $H^{-1}$ is also symmetric-positive definite:

$$H \overset{(30)}{=} 2 X X^\top = 2(X X^\top)^\top = H^\top, \tag{32}$$

$$H^{-1} = (2 X X^\top)^{-1} \overset{(32)}{=} \left( (2 X X^\top)^\top \right)^{-1} = \left( (2 X X^\top)^{-1} \right)^\top = (H^{-1})^\top. \tag{33}$$

The first term in (28) is equal to zero since for $0 \in \mathbb{R}^{1 \times b}$, $g(0) = \| 0 X \|_2^2 = 0$. We assume that weights of the network are pre-trained well enough, so we are at a local optimum (or near the local optimum), where $\frac{\partial g}{\partial \delta}(0) = 0$. Hence, the second term can also be ignored. The last term approaching zero when the change of weights $\delta$ is small, so finally we have

$$g(\delta) \approx \frac{1}{2} \delta H \delta^\top. \tag{34}$$

Now we aim to find such $\delta$ that minimize the loss (34) and eliminates the $q^{\text{th}}$ component of the weights vector $w$:

$$\min_{\delta \in \mathbb{R}^{1 \times b}} \frac{1}{2} \delta H \delta^\top, \tag{35}$$

$$\text{subject to} \quad \delta e^q + w_q = 0.$$

Then the Lagrangian function will be

$$L(\delta, \lambda) := \frac{1}{2} \delta H \delta^\top + \lambda \left( \delta e^q + w_q \right). \tag{36}$$

Let $\delta^\star$ and $\lambda^\star$ represent the values of $\delta$ and $\lambda$ that minimize the Lagrangian function given in Equation 36. To find these optimal values, we can take the partial derivatives of the Lagrangian $L$ with respect to $\delta$ and $\lambda$, respectively. The solution is then obtained by identifying the points at which these partial derivatives are equal to zero:

$$\frac{\partial L}{\partial \delta}(\delta^\star, \lambda^\star) = 0, \quad \frac{\partial L}{\partial \lambda}(\delta^\star, \lambda^\star) = 0.$$

These conditions provide the necessary criteria for identifying the optimal values $\delta^\star$ and $\lambda^\star$, that minimize the Lagrangian:

$$\frac{\partial L}{\partial \delta} = H\delta^{\star\top} + \lambda^\star e^q = 0, \tag{37}$$

$$\frac{\partial L}{\partial \lambda} = \delta^\star e^q + w_q = 0. \tag{38}$$

Let us assume the Hessian matrix $H$ to be invertible. From (37) we have $\delta^\star = -\lambda^\star(H^{-1}e^q)^\top = -\lambda^\star(H_{:q}^{-1})^\top \overset{(33)}{=} -\lambda^\star H_{q:}^{-1}$, where the last equality holds because $H^{-1}$ is symmetric positive-definite matrix. Substituting this into (38) gives us $-\lambda^\star e_q^\top H^{-1} e^q + w_q = 0$, hence $\lambda^\star = \frac{w_q}{H_{qq}^{-1}}$. Now we can substitute this result into (37) to obtain $H\delta^{\star\top} + \frac{w_q}{H_{qq}^{-1}} e^q = 0$, hence we have

$$\delta^\star = -\frac{w_q}{H_{qq}^{-1}} H_{q:}^{-1}. \tag{39}$$

Let us denote as $S_q^{\text{OBS}}$ to be the optimal value of Lagrangian function (36) with optimal $\delta^\star$ from (39), we will refer to this value as a pruning metric or saliency metric:

$$S_q^{\text{OBS}} := L(\delta^\star, \lambda^\star) \overset{(36)}{=} \frac{1}{2}\left(\frac{w_q}{H_{qq}^{-1}}\right)^2 (e^q)^\top H^{-1} H H^{-1} e^q = \frac{1}{2}\frac{w_q^2}{H_{qq}^{-1}}. \tag{40}$$

Now we have to find such an index $q$ that gives us the smallest value of $S_q^{\text{OBS}} = \frac{1}{2}\frac{w_q^2}{H_{qq}^{-1}}$.

Recall that we aim to solve the problem (21). Let us denote $\Delta^\star$ as the optimal change of weights for the whole matrix $W$. In addition we denote $S_{kq}^{\text{OBS}}$ as a pruning metric (saliency metric) for the whole matrix $W$. Hence, we have to find the best row $k$ and best index $q$ such as $S_{kq}^{\text{OBS}}$ will be minimal and then update the $k^{\text{th}}$ row by using the rule (39):

$$\Delta_{k:}^\star = -\frac{W_{kq}}{H_{qq}^{-1}} H_{q:}^{-1}, \qquad \Delta_{i:}^\star = 0 \in \mathbb{R}^{1\times b} \text{ for } i \neq k, \qquad S_{kq}^{\text{OBS}} := \frac{1}{2}\frac{W_{kq}^2}{H_{qq}^{-1}}. \tag{41}$$

That was the method of pruning for neural networks described in Optimal Brain Surgeon paper.

Optimal Brain Damage assumed that the Hessian $H$ is a diagonal matrix, which makes it very simple to find the inverse of it. So the pruning metric from (41) simplifies to

$$S_{kq}^{\text{OBD}} = \frac{1}{2}\frac{W_{kq}^2}{H_{qq}^{-1}} = \frac{1}{2}\frac{W_{kq}^2}{(H_{qq})^{-1}} = W_{kq}^2 \sum_{i=1}^{a} X_{qi}^2 = W_{kq}^2 \|X_{q:}\|_2^2 = (|W_{kq}|\|X_{q:}\|_2)^2, \tag{42}$$

where $X_{q:} \in \mathbb{R}^{1\times a}$ is the $q^{\text{th}}$ row of the matrix $X$.

### G.3 SPARSEGPT

The main problem of OBD and OBS is that after you prune the first parameter $W_{kq}$, you should not consider this parameter for pruning on the next step. Let us denote as $W_{k:}' \in \mathbb{R}^{1\times(b-1)}$ to be the set of parameters eligible for consideration for the $k^{\text{th}}$ row of $W$ on the second step of pruning, then

$$W_{k:}' := \begin{pmatrix} W_{k,1:q-1} & W_{k,q+1:b} \end{pmatrix},$$

where

$$W_{k,j_1:j_2} := \begin{pmatrix} W_{ij_1} \cdots W_{ij_2} \end{pmatrix} \in \mathbb{R}^{1\times(j_2-j_1)},$$

for any $1 \leq j_1 \leq j_2 \leq b$.

We also denote as $X' \in \mathbb{R}^{(b-1)\times a}$ to be matrix $X$ without the $q^{\text{th}}$ row:

$$X' := \begin{pmatrix} X_{1:}^\top & \cdots & X_{q-1:}^\top & X_{q+1:}^\top & \cdots & X_{b:}^\top \end{pmatrix}^\top.$$

Finally, we define $g' : \mathbb{R}^{b-1} \to \mathbb{R}$ in the following way:

$$g'(\delta') := \|\delta' X'\|_2^2. \tag{43}$$

The Hessian for the $k^{\text{th}}$ row will not be the same as Hessian for all other rows, moreover it will have different size. Indeed, after we pruned the parameter $W_{kq}$, the Hessians will be:

$$H'_{i \neq k} := \frac{\partial^2}{\partial W_{i:}} g(W_{i:}) \overset{(30)}{=} 2XX^\top \in \mathbb{R}^{b \times b} \quad \text{for } i \neq k,$$

$$H'_k := \frac{\partial^2}{\partial W'_{k:}} g'(W'_{k:}) \overset{(43)}{=} 2X'X'^\top \in \mathbb{R}^{(b-1) \times (b-1)}.$$

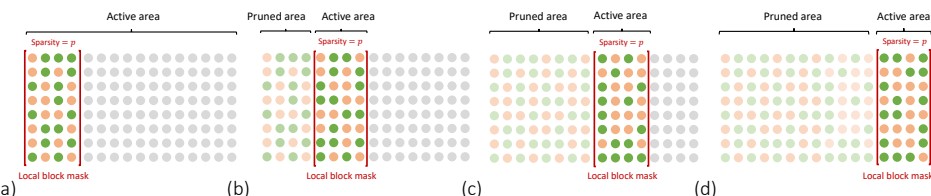

(a)     (b)     (c)     (d)

Figure 3: Demonstration of SparseGPT mask selection algorithm. Entries of $M$ that are equal to one are represented by orange circles, while zeros are depicted with green circles. In the beginning (a), we prune the first block of parameters. To do this, we compute the mask for the local block of size $B$. The local block should be $p\%$ sparse, so the local block mask will be $p\%$ dense. On the second step (b) we compute the mask for the second local block of the same size with the same local sparsity $p\%$ and so on.

The situation will be even more complex as more parameters of the model will be pruned because eventually you will have to compute $c$ different Hessians of masked parameters for each row. One such Hessian requires $O(b^3)$ time to compute. This results in overall complexity of $O(cb^3)$.

This problem can be solved by constraining each row to have the same pruning mask resulting in a structured sparsity mask, but such a constraint would greatly influence on the accuracy of the model. Is is known that ResNet50 can be pruned to $p = 0.9$ unstructured sparsity without big degradation of quality by (Evci et al., 2020) or (Peste et al., 2021) , whereas structured pruning to $p = 0.5$ is challenging even with fine tuning (Liu et al., 2021).

SparseGPT (Frantar & Alistarh, 2023) solves this problem by pruning parameters one by one in a sequential order from left to right. SparseGPT consider only unseen parameters on the right side of the sequence. By doing that we make sure that Hessian is the same for all rows like it was on the first step of pruning.

Let us consider the core idea of the SparseGPT algorithm (Algorithm 5). Initially, the algorithm takes a weight matrix $W$ and an input matrix $X$ along with block sizes $B$ and $B_s$ and a sparsity ratio $p \in [0, 1)$, which defines how much of the model will be pruned. It begins by setting an adaptive mask matrix, $M$, with all entries initialized to $0$, indicating that no weights are pruned initially. Additionally, it computes full Hessian matrix $H$.

The algorithm proceeds through the weight matrix in blocks, determined by the block size parameter $B$. For each outer iteration, it defines a range of columns from the weight matrix to process. As it steps through these columns, it checks if a specific column index aligns with the adaptive block size, $B_s$. As for now let us assume $B_s = B$, so we define the local mask for a full block of size $B$. Specifically, it selects $p\%$ of the weights within that block that have the smallest value of $S_{kq}^{\text{OBD}} := W_{kq}^2 / H_{qq}^{-1}$. The demonstration of SparseGPT mask selection algorithm you can see on the Figure 3.

The dynamic mask selection mechanism, illustrated in Figure 3, allows the pruning mask to adapt based on changes that occur as parameters are removed. This adaptability ensures that the mask reflects the evolving importance of the remaining parameters throughout the pruning process. In contrast, applying a fixed global mask to the entire matrix would lead to suboptimal results. The

reason is that parameter updates can shift the relative importance of weights some parameters initially deemed unimportant may become critical after others are pruned, while previously significant parameters may lose their relevance. Thus, a dynamic approach is essential to achieve better final accuracy by continuously adjusting the pruning decisions to the changing landscape of the model's internal structure.

Once the mask is updated for the current block, the algorithm moves to the next step: updating the weights. It scans through each row of the selected columns and checks if the corresponding entry in the mask should be pruned. If the mask allows the weight to be removed, the algorithm adjusts the weight values from the row using an update rule (41) from OBS (LeCun et al., 1989) that leverages the inverse Hessian matrix. This rule ensures that the remaining weights are adjusted (off-block parameters communication) to compensate for the loss of pruned weights, thereby minimizing the error (12) introduced by pruning. The demonstration of what parameters should be updated are on the Figure 4.

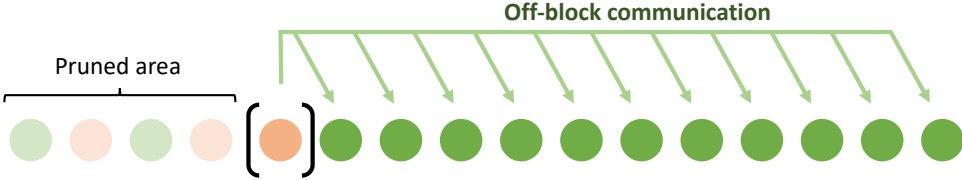

Figure 4: Demonstration of off-block parameters communication for SparseGPT. Entries of $W$ marked with orange circles indicate parameters designated for removal, while those marked with green circles represent weights that require updating. When we prune the parameter, all parameters on the right part of the sequence are updated.

After processing the current block, the algorithm modifies the Hessian matrix by removing its first row and column, effectively reducing the problem size for subsequent iterations. This update reflects the incremental pruning process, ensuring that the inverse Hessian matrix remains aligned with the remaining active weights.

The algorithm continues iterating over the weight matrix in blocks, applying the same procedure: updating the mask, adjusting the remaining weights, and refining the Hessian matrix. This iterative approach allows it to prune the model in a structured way, ensuring that even though many weights are removed, the performance loss is minimized by carefully selecting which weights to prune and by adapting the remaining ones column by column. The end result is a sparse version of the original model, optimized for reduced computational load while preserving as much accuracy as possible.

---

**Algorithm 5** SparseGPT pruning algorithm

---

1: **Initialization:** Input matrix $X \in \mathbb{R}^{b \times a}$, weight matrix $W \in \mathbb{R}^{c \times b}$, block size for parameters update $B$, adaptive mask block size $B_s$, sparsity ratio $p \in [0, 1)$.
2: $M \leftarrow 0 \in \mathbb{R}^{c \times b}$
3: $H \leftarrow 2XX^\top + \lambda I$
4: **for** $j_1 = 0, \ B, \ 2B, \ \cdots$ **do**
5:     **for** $j_2 = j_1, \cdots, j_1 + B$ **do**
6:         **if** $j_2 \bmod B_s = 0$ **then**
7:             $M_{:,j_2:j_2+B_s} \leftarrow$ mask of $p\%$ weights $W_{kq}$ from the matrix $W_{:,j_2:j_2+B_s}$ with smallest $W_{kq}^2/[H^{-1}]_{qq}$
8:         **end if**
9:         **for** $i = 1, \cdots, c$ **do**
10:           **if** $M_{ij_2} = 1$ **then**
11:             $W_{i,j_2:b} \leftarrow W_{i,j_2:b} - \frac{W_{i,j_2}}{H_{j_2 j_2}^{-1}} H_{j_2:}^{-1}$
12:           **end if**
13:         **end for**
14:         $H \leftarrow H_{2:,2:}$
15:     **end for**
16: **end for**

---

SparseGPT can be seamlessly extended to support semi-structured pruning patterns, including the widely-used $n : m$ sparsity format (Zhou et al., 2021; Hubara et al., 2021a; Mishra et al., 2021), which offers speedups on NVIDIA Ampere GPUs with its 2:4 implementation (Lin et al., 2023; NVIDIA Corporation, 2020). In this format, every block of $m$ consecutive weights must contain exactly $n$ zeros. To implement this, we set the block size $B_s = m$ and ensure that the pruning mask for each row retains the $n$ weights that minimize the error. This approach can be generalized to other semi-structured pruning patterns. However, increasing $B_s$ would be ineffective, as it would prevent a balanced distribution of zeros across different column groups of size $m$.

In the real implementation of SparseGPT several techniques are used to reduce the number of computational steps. For example, we can avoid computing the inverse of the updated Hessian after pruning of any column by using the Cholesky decomposition and perform the update rule for all rows at once by using numpy broadcasting.

We can estimate the total complexity for pruning of one layer of LLM. We have to make the following steps to apply SparseGPT:

- Compute $H = 2XX^\top$ in $\mathcal{O}(ab^2)$ and its Cholesky decomposition: $\mathcal{O}(b^3)$
- Calculate pruning metric $S_{kq}^{\text{OBS}}$ (41) and sort it for all blocks in $\mathcal{O}(cb \log c)$
- Calculate the update rule for every weight in $\mathcal{O}(cb^2)$

Total complexity is
$$T_{\text{SparseGPT}} = \mathcal{O}\left(ab^2 + b^3 + cb^2 + cb \log c\right) \tag{44}$$

## G.4 WANDA

---
**Algorithm 6** Wanda pruning algorithm

---
1: **Initialization:** Input matrix $X \in \mathbb{R}^{b \times a}$, weight matrix $W \in \mathbb{R}^{c \times b}$, sparsity ratio $p \in [0, 1)$.
2: $M \leftarrow 0 \in \mathbb{R}^{c \times b}$
3: **for** $i = 1, \cdots, c$ **do**
4:     $M_{i:} \leftarrow$ mask of $p\%$ weights $W_{iq}$ from a row-vector $W_{i:}$ with smallest $|W_{iq}|\|X_{q:}\|_2$
5: **end for**
6: $W \leftarrow W - M \odot W$

---

Wanda (Sun et al., 2023) is trying to simplify the intensive computation of the inverse of the Hessian by assuming the Hessian to be diagonal. So, the relationship between OBD and OBS is the same as the relationship between Wanda and SparseGPT (Frantar & Alistarh, 2023). Hence the pruning metric (41) simplifies to

$$S_{kq}^{\text{OBD}} = \frac{1}{2}\frac{W_{kq}^2}{H_{qq}^{-1}} = \frac{1}{2}\frac{W_{kq}^2}{(H_{qq})^{-1}} = W_{kq}^2 \sum_{i=1}^{a} X_{qi}^2 = W_{kq}^2 \|X_{q:}\|_2^2 = \left(|W_{kq}|\|X_{q:}\|_2\right)^2 ,$$

where $X_{q:} \in \mathbb{R}^{1 \times a}$ is the $q^{\text{th}}$ row of the matrix $X$.

We define a mapping $\psi_X : \mathbb{R}^{c \times b} \to \{0, 1\}^{c \times b}$, which operates on a portion of the weight matrix $W$ and produces a pruning mask. For each element $W_{ij}$, this mapping calculates the pruning metric $|W_{ij}|\|X_{j:}\|_2$. It then returns a mask that contains ones at the positions corresponding to the $r$ smallest values of this metric:

$$\psi_X(W, r) := \text{mask of the } r \text{ weights } W_{ij} \text{ with the smallest } |W_{ij}|\|X_{j:}\|_2 \text{ values.} \tag{45}$$

For example, let us have a weight matrix
$$W = \begin{pmatrix} 3 & -2 \\ -2 & 4 \\ 1 & -6 \end{pmatrix},$$

and an input matrix
$$X = \begin{pmatrix} 4 & 3 \\ 0 & 1 \end{pmatrix},$$

now we compute the metric $|W_{ij}|\|X_{j:}\|_2$ for every entry of $W$:

$$\begin{pmatrix} |W_{11}|\|X_{1:}\|_2 & |W_{12}|\|X_{2:}\|_2 \\ |W_{21}|\|X_{1:}\|_2 & |W_{22}|\|X_{2:}\|_2 \\ |W_{31}|\|X_{1:}\|_2 & |W_{32}|\|X_{2:}\|_2 \end{pmatrix} = \begin{pmatrix} 3 \cdot 5 & 2 \cdot 1 \\ 2 \cdot 5 & 4 \cdot 1 \\ 1 \cdot 5 & 6 \cdot 1 \end{pmatrix} = \begin{pmatrix} 15 & 2 \\ 10 & 4 \\ 5 & 6 \end{pmatrix},$$

then

$$\psi_X(W, 1) = \begin{pmatrix} 0 & 1 \\ 0 & 0 \\ 0 & 0 \end{pmatrix}, \quad \psi_X(W, 2) = \begin{pmatrix} 0 & 1 \\ 0 & 1 \\ 0 & 0 \end{pmatrix}, \quad \psi_X(W, 4) = \begin{pmatrix} 0 & 1 \\ 0 & 1 \\ 1 & 1 \end{pmatrix}.$$

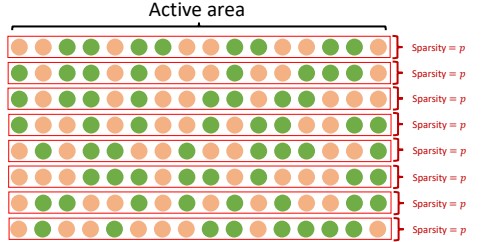
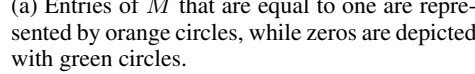
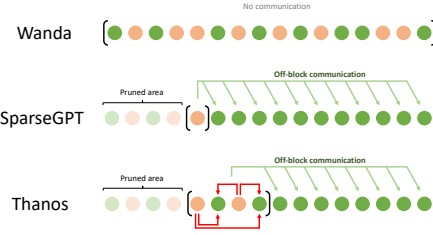

(a) Entries of $M$ that are equal to one are represented by orange circles, while zeros are depicted with green circles.

(b) Entries of $W$ marked with orange circles represent parameters that should to be removed. In Wanda we do not have parameters update.

Figure 5: Demonstration of (a) Wanda mask selection algorithm and (b) comparison of inter-parameters communication of several pruning methods.

The Wanda pruning algorithm (Algorithm 6) offers a structured way to prune a model's weight matrix by leveraging both the weight magnitudes and input norms. Its objective is to remove a specific percentage of the least significant weights while maintaining the model's performance.

The algorithm takes as input the matrix of weights $W$ and the input matrix $X$, along with a sparsity ratio $p \in [0, 1)$ that dictates what fraction of the weights will be pruned. It also initializes a mask matrix $M$ with all entries set to zero, which will be updated to indicate which weights are to be pruned.

The core of the algorithm revolves around processing the weight matrix row by row. For each row, the goal is to identify a subset of weights that contribute the least to the overall computation. Specifically, the algorithm examines the magnitude of each weight in the row along with the norm of the corresponding input column. The product of the absolute weight value and the $l^2$-norm of the input column serves as a metric for determining the importance of each weight (42).

Once the metric is computed, the algorithm selects the least important weights from that row. The number of weights to prune is determined by the sparsity ratio $p$. The mask matrix $M$ is then updated, marking the positions of the selected weights with ones, indicating that these weights will be removed. Note, that Wanda forces each row of $M$ to be $p\%$ dense (Figure 5a). The entire mask for the weight matrix $W$ is constructed in a single step, without iterative adjustments or recalculations

After processing all rows of the weight matrix, the algorithm applies the pruning step. It multiplies the mask matrix element-wise with the weight matrix, effectively setting the selected weights to zero. The final step subtracts the masked weights from the original weight matrix, ensuring that the pruned weights are removed while the remaining weights are preserved. In contrast to the Algorithm 5, in Wanda we do not have weights adjustment (Figure 5b). In Wanda, parameters remain unchanged after pruning, focusing solely on selecting and removing weights. In contrast, both SparseGPT and Thanos incorporate parameter updates following the pruning process.

The Wanda algorithm produces a sparse version of the original weight matrix. By carefully selecting which weights to prune based on both their magnitudes and the corresponding input norms, the algorithm ensures that the pruning process minimally impacts the model's performance. This approach results in a more compact and efficient model, with unimportant weights removed while retaining the key ones that contribute most to the output.

To apply Wanda we have to follow these steps:

- Calculate pruning metric (42) $S_{kq}^{\text{OBD}}$ for all possible choice of $k$ and $q$ and find $p\%$ smallest values for every row in $\mathcal{O}(ab + cb \log b)$

- Calculate the update in $\mathcal{O}(1)$ and prune weights in $\mathcal{O}(cb)$

Total complexity is

$$T_{\text{Wanda}} = \mathcal{O}\left(ab + cb \log b\right). \tag{46}$$

## H  THANOS PRUNING

SparseGPT (Frantar & Alistarh, 2023) algorithm (Algorithm 5) prunes weights column by column, removing only one weight from each row at a time. On the other hand, we may benefit from updating more weights at once.

### H.1  UPDATING SEVERAL WEIGHTS AT A TIME

Assuming we aim to prune $s$ weights at a time, the problem becomes analogous to (35), but now includes $s$ constraints. Specifically, the weights $w_{q_1}, \cdots, w_{q_s}$ must be removed, where $1 \leq q_1 < \cdots < q_s \leq b$ represent the indices of the weights selected for pruning. This modifies the optimization problem, which can now be expressed as follows:

$$\min_{\delta \in \mathbb{R}^{1 \times b}} \frac{1}{2} \delta H \delta^\top, \tag{47}$$

$$\text{subject to}$$

$$\delta e^{q_1} + w_{q_1} = 0,$$

$$\vdots$$

$$\delta e^{q_s} + w_{q_s} = 0.$$

The Lagrangian function for the problem (47) will have the following form:

$$\widehat{L}(\delta, \lambda_1, \cdots, \lambda_s) := \frac{1}{2} \delta H \delta^\top + \sum_{j=1}^{s} \lambda_j \left(\delta e^{q_j} + w_{q_j}\right). \tag{48}$$

Let $\widehat{\delta}$ and $\widehat{\lambda}_1, \cdots, \widehat{\lambda}_s$ represent the values of $\delta$ and $\lambda_1, \cdots, \lambda_s$ that minimize the Lagrangian function given in Equation 48. To find these optimal values, we can take the partial derivatives of the Lagrangian $\widehat{L}$ with respect to $\delta$ and $\lambda_1, \cdots, \lambda_s$, respectively. The solution is then obtained by identifying the points at which these partial derivatives are equal to zero:

$$\frac{\partial \widehat{L}}{\partial \delta}(\widehat{\delta}, \widehat{\lambda}_1, \cdots, \widehat{\lambda}_s) = 0, \quad \frac{\partial \widehat{L}}{\partial \lambda_1}(\widehat{\delta}, \widehat{\lambda}_1, \cdots, \widehat{\lambda}_s) = 0, \quad \cdots, \quad \frac{\partial \widehat{L}}{\partial \lambda_s}(\widehat{\delta}, \widehat{\lambda}_1, \cdots, \widehat{\lambda}_s) = 0.$$

These conditions provide the necessary criteria for identifying the optimal values $\widehat{\delta}$ and $\widehat{\lambda}_1, \cdots, \widehat{\lambda}_s$, that minimize the Lagrangian:

$$\frac{\partial \widehat{L}}{\partial \delta} = H \widehat{\delta}^\top + \sum_{j=1}^{s} \widehat{\lambda}_j e^{q_j} = 0, \tag{49}$$

$$\frac{\partial \widehat{L}}{\partial \lambda_1} = \widehat{\delta} e^{q_1} + w_{q_1} = 0,$$

$$\vdots \tag{50}$$

$$\frac{\partial \widehat{L}}{\partial \lambda_s} = \widehat{\delta} e^{q_s} + w_{q_s} = 0.$$

Now we can express $\widehat{\delta}$ as a linear combination of $\widehat{\lambda}_1, \cdots, \widehat{\lambda}_s$:

$$\widehat{\delta} \stackrel{(49)}{=} -\sum_{j=1}^{s} \widehat{\lambda}_j H_{q_j:}^{-1} = -\widehat{\lambda}R, \tag{51}$$

where $\widehat{\lambda} := \left( \widehat{\lambda}_1 \ \cdots \ \widehat{\lambda}_s \right) \in \mathbb{R}^{1 \times s}$ and

$$R := \begin{pmatrix} H_{q_1:}^{-1} \\ \vdots \\ H_{q_s:}^{-1} \end{pmatrix} \in \mathbb{R}^{s \times b}. \tag{52}$$

By substituting (51) into (50) we can find $\widehat{\lambda}$:

$$-\widehat{\lambda}Re^{q_1} + w_{q_1} = 0,$$

$$\vdots$$

$$-\widehat{\lambda}Re^{q_s} + w_{q_s} = 0.$$

We can combine all $s$ equations and write it in a matrix form:

$$\widehat{\lambda}\widehat{R} = u, \tag{53}$$

where

$$\widehat{R} := (R_{:q_1} \ \cdots \ R_{:q_s}) \in \mathbb{R}^{s \times s} \tag{54}$$

is a square matrix and

$$u := (w_{q_1} \ \cdots \ w_{q_s}) \in \mathbb{R}^{1 \times s}. \tag{55}$$

Hence, we need to solve the system of linear equation (53) to find $\widehat{\lambda}$. Then we can use Lagrangian multipliers $\widehat{\lambda}_1, \cdots, \widehat{\lambda}_s$ to get the optimal change of weights:

$$\widehat{\delta} \stackrel{(53) \to (51)}{=} -u\widehat{R}^{-1}R. \tag{56}$$

Let us denote as $S$ to be the optimal value of Lagrangian function (48) with optimal $\widehat{\delta}$ from (56), then:

$$S := \widehat{L}(\widehat{\delta}, \widehat{\lambda}_1, \cdots, \widehat{\lambda}_s) \stackrel{(56) \to (48)}{=} \frac{1}{2}u\widehat{R}^{-1}RHR^{\top}(\widehat{R}^{-1})^{\top}u^{\top}. \tag{57}$$

The main challenge with this approach lies in its computational complexity. To determine the optimal indices $q_1, \cdots, q_s$ for pruning, we face a combinatorial explosion, as there are $\binom{b}{s} = \frac{b!}{s!(b-s)!}$ possible combinations for each row. For each combination of $q_1, \cdots, q_s$, we need to compute $\widehat{\lambda}, \widehat{\delta}$, and $S$, which requires $\mathcal{O}\left(\frac{b!}{s!(b-s)!}\left(s^3 + c^2 + cs\right)\right)$ operations. This makes the approach impractical for LLMs.

The problem becomes substantially more manageable if the pruning mask is predefined, as this removes the need to search for the optimal weights to prune. Instead, the focus shifts to optimizing the weight adjustments for the already selected parameters, simplifying the task significantly. This approach effectively divides the problem into two steps: first, identifying the best pruning mask, and then determining the optimal weight adjustments for the parameters within this mask.

### H.2 Pruning in a block-wise manner

Obtaining the optimal weight adjustments as defined in Equation (56) for the entire matrix $W \in \mathbb{R}^{c \times b}$ can be challenging due to the high computational cost associated with solving large systems of linear equations. To address this, we propose dividing $W$ into a reasonable number of smaller blocks, each of which is sized to allow feasible computation. Notably, SparseGPT (Algorithm 5) has demonstrated the effectiveness of dividing the weight matrix $W$ into blocks, as this approach allows for dynamic updating of the pruning mask. This adaptability is valuable, as weights can change

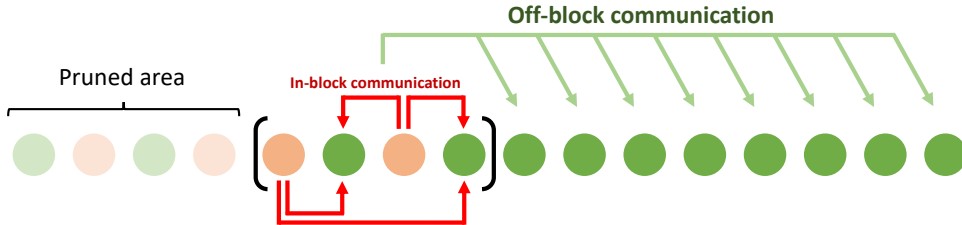

Figure 6: Demonstration of in-block and off-block parameters communication for Thanos. Entries of $W$ marked with orange circles indicate parameters designated for removal, while those marked with green circles represent weights that require updating. When pruning a block of parameters of size $B$, all parameters to the right of the block within the sequence are updated, a process referred to as off-block communication. Additionally, updates occur within the parameters inside the pruning block itself, known as in-block communication.

significantly during the pruning process, making it advantageous to adjust the mask accordingly as pruning progresses.

Let us denote $B \in \{1, \cdots, b\}$ as the number of parameters we update at once while pruning by Thanos. We will call this parameter as the block size. With this approach, the matrix $W$ is partitioned into $\lceil \frac{b}{B} \rceil$ blocks, referred to as local blocks, where $\lceil \cdot \rceil : \mathbb{R} \to \mathbb{Z}$ is a ceiling function, defined by

$$\lceil x \rceil := \min \{n \in \mathbb{Z} \mid n \geq x\}.$$

By reducing the number of weights to be pruned within each row of a block, we make the problem more computationally manageable. The pruning algorithm then iterates through each local block in $W$, starting from the leftmost block. For each block, it solves the system of linear equations for every row and applies the update rule from Equation (56) to all remaining weights located to the right of the current block in the sequence.

Let's formalize the process described above. For the pruning procedure, we define a sequence of matrices $\{\widehat{W}^0, \ldots, \widehat{W}^{\lfloor b/B \rfloor}\}$, starting with the following initial state:

$$\widehat{W}^0 = W. \tag{58}$$

This sequence is then computed iteratively for each $j \in \{0, \ldots, \lfloor b/B \rfloor\}$ according to the following rule:

$$\widehat{W}^{j+1} = \left(\widehat{W}^j + \widehat{\Delta}^j\right)_{:,B+1:}, \tag{59}$$

where $A_{:,B+1:}$ represents the matrix $A$ with the first $B$ columns removed, and $\widehat{\Delta}^j \in \mathbb{R}^{c \times (b-jB)}$ is an update rule applied at each step. In essence, each new matrix $\widehat{W}^{j+1}$ is obtained by summing the previous matrix $\widehat{W}^j$ with the update $\widehat{\Delta}^j$, followed by the removal of the first $B$ columns from the result.

At each $j^{\text{th}}$ pruning step, we construct a local block mask to identify parameters for pruning from the first $B$ columns of $\widehat{W}^j$. (Further details on the construction of the pruning mask are available in Section H.4.) Using this mask, we determine the indices of parameters to be removed for each row of $\widehat{W}^j$. We then prune the parameters within the block and compute the update rule $\widehat{\delta}$ for each row of $\widehat{W}^j$ according to Equation (56):

$$\widehat{\Delta}^j := \begin{bmatrix} \widehat{\delta}^{j1} \\ \vdots \\ \widehat{\delta}^{jc} \end{bmatrix} \in \mathbb{R}^{c \times (b-jB)}, \tag{60}$$

where $\widehat{\delta}^{ji} \in \mathbb{R}^{1 \times (b-jB)}$ represents the update rule derived from Equation (56) for the $i^{\text{th}}$ row of $\widehat{W}^j$.

During this process, only the parameters within the local block itself are removed. However, the parameters of other blocks may still be influenced, as the update rule (56) applies to weights outside the current block. This creates what we term in-block parameter communication, where parameters within a block communicate through the update rule to reflect the changes resulting from pruned weights in the same block.

Additionally, there is off-block parameter communication. This refers to how pruned parameters within a block influence the weights beyond the block, or in the sequence of remaining parameters on the right, via the update rule (56). These interactions, both in-block and off-block, ensure that the pruning effects are efficiently propagated throughout the matrix (Figure 6).

To keep the computational cost reasonable, it is important to select $B \in \{1, \cdots, b\}$ so that $B$ remains small enough to avoid high costs, but hight enough to gain more out of in-block communication. This is crucial because for each row of the weight matrix $W$, solving the linear system requires $\mathcal{O}(B^3)$ operations.

### H.3 PRUNING METRIC

Thanos employs the same pruning metric as Wanda. It can be readily shown that Wanda provides an optimal solution when the goal is to remove a single weight at a time without tuning the remaining weights.

To see this, consider the problem in which we aim to remove one weight from $W \in \mathbb{R}^{c \times b}$ while keeping the loss as low as possible, under the constraint that no other weights in the row can be adjusted. Hence, our task is to remove a single weight with not parameters tuning:

$$\min_{\delta \in \mathbb{R}^{1 \times b}} \|\delta X\|_2^2, \tag{61}$$

$$\text{subject to}$$

$$\delta e^q + W_{kq} = 0,$$

$$\delta_j = 0 \text{ for all } j \neq q.$$

To address this problem, we evaluate the loss (Equation 61) for each possible combination of $k \in \{1, \cdots, c\}$ and $q \in \{1, \cdots, b\}$, selecting the values that yield the minimum loss.

This loss computation can be performed with high efficiency. Specifically, when removing a single parameter without adjusting others, the loss calculation simplifies considerably due to the localized impact of removing just one weight. This allows us to compute the resulting loss incrementally, leveraging the fixed values of the remaining weights to isolate the effect of the removal on the overall loss. Consequently, this approach minimizes computational overhead while ensuring that the selected parameter yields the smallest possible loss increase.

$$\min_{\delta \in \mathbb{R}^{1 \times b}} \|\delta X\|_2^2 = \min_{q \in \{1, \cdots, b\}} \left\| [0 \cdots w_q \cdots 0] \begin{bmatrix} x_{11} & \cdots & x_{1a} \\ \vdots & & \vdots \\ x_{q1} & \cdots & x_{qa} \\ \vdots & & \vdots \\ x_{b1} & \cdots & x_{ba} \end{bmatrix} \right\|_2^2 =$$

$$= \min_{q \in \{1, \cdots, b\}} \left\| w_q \begin{bmatrix} x_{q1} \\ \vdots \\ x_{qa} \end{bmatrix} \right\|_2^2 = \min_{q \in \{1, \cdots, b\}} w_q^2 \|X_{q:}\|_2^2. \tag{62}$$

The metric in Equation (H.3) aligns precisely with the Wanda metric, making Wanda the optimal solution for the problem formulated in (61).

In contrast, the SparseGPT metric (Equation 41) provides a locally optimal solution when the goal is to remove a single parameter, with the flexibility to adjust the remaining parameters. However, this approach becomes problematic when scaling to multiple parameters. While SparseGPT excels under

the constraint of single-parameter removal with a specific weight update rule defined in Equation (41), Thanos prunes multiple weights simultaneously and employs its own update rule (Equation 56). Consequently, the solutions derived from SparseGPT are not guaranteed to be optimal for cases where multiple parameters are pruned together.

On the other hand, the sequential application of the Wanda metric ensures optimality as long as the weights remain unchanged during the pruning process. Even though Wanda does not modify weights throughout the process, we can still perform weight updates after the pruning step, resulting in a solution that is incrementally closer to optimal. This combination of sequential pruning followed by weight adjustment enables the model to achieve better overall performance than would be possible with a static, single-step pruning approach.

### H.4 PRUNING MASK SELECTION

When removing weights, we first construct some local part of the mask $M \in \mathbb{R}^{c \times b}$ to identify which weights should be pruned. This mask is essential for setting up and solving the systems of linear equations in Equations (49) and (50). In this chapter, we detail the process of constructing this mask and demonstrate how it is applied to achieve the optimal weight adjustments defined in Equation (56). We cover the methodology for both unstructured (H.4.1) and structured (H.4.5) sparsity configurations.

### H.4.1 UNSTRUCTURED SPARSITY

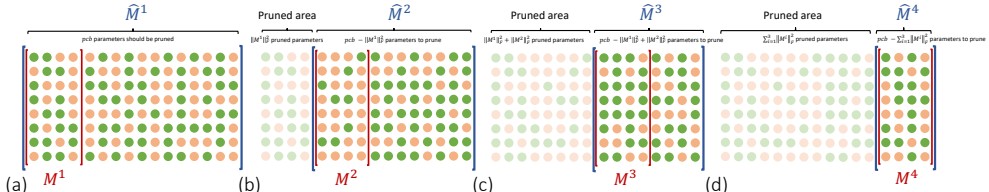

Figure 7: Demonstration of Thanos mask selection algorithm. Entries of $M$ that are equal to one are represented by orange circles, while zeros are depicted with green circles. In the beginning (a), we prune the first block of parameters. To do this, we compute the mask for the whole matrix and use only local part of it. On the second step (b) we compute the mask for the global residual matrix (blue colored), then select a local part of it and so on.

SparseGPT (Frantar & Alistarh, 2023) uses local information in each subsequent block to construct a pruning mask. Each block should have the same sparsity ratio and hence the whole matrix will have the same overall sparsity (Figure 3). This approach is efficient because it allows us to take into account changing of weights during pruning: when we prune the block, we change all weights coming after the block and hence the pruning mask can significantly change. On the other hand, such a local mask selection can limit the efficiency of the pruning because we constraint each block to have a fixed sparsity ratio.

The counterpart of SparseGPT – Wanda (Sun et al., 2023), constructs the global mask with row sparsity constraints: each row should have the same sparsity (Figure 5a).

We present the algorithm of mask selection that is not limited to local block or local row sparsity and allows us to dynamically adjust to weights update.

Let us have a weight matrix $W \in \mathbb{R}^{c \times b}$, then we aim to construct the pruning mask $M \in \{0, 1\}^{c \times b}$ that contains ones for entries of $W$ that should be removed. The desired sparsity ratio is $p \in [0, 1]$. The final weights matrix $\widehat{W} \in \mathbb{R}^{c \times b}$ should have $p\%$ sparsity level. Hence, the mask should be $p\%$ dense. In total we have to remove $\lceil pcb \rceil$ parameters from the matrix $W$, so

$$\|M\|_F^2 \stackrel{(13)}{=} \lfloor pcb \rfloor.$$

We construct the mask $M$ and the final weight matrix $\widehat{W}$ iteratively, generating sequences of masks $\{M^0, M^1, \ldots, M^{\lfloor \frac{b}{B} \rfloor}\}$ and weights $\{\widehat{W}^0, \widehat{W}^1, \ldots, \widehat{W}^{\lfloor \frac{b}{B} \rfloor}\}$, starting from the initial values:

$$M^0 = 0 \in \mathbb{R}^{c \times b}, \tag{63}$$

$$\widehat{W}^0 \stackrel{(58)}{=} W. \tag{64}$$

We further introduce a sequence of global residual masks $\{\widehat{M}^1, \ldots, \widehat{M}^{\lfloor \frac{b}{B} \rfloor}\}$, which assist in constructing each $M^k$. The sequences are constructed iteratively as follows: for all $j \in \{0, \cdots, \lfloor \frac{b}{B} \rfloor\}$

$$\widehat{M}^{j+1} \stackrel{(45)}{=} \psi_X \left( \widehat{W}^j, \; pcb - \sum_{i=1}^{j} \|M^i\|_F^2 \right), \tag{65}$$

$$M^{j+1} = \widehat{M}^{j+1}_{:,1:B}, \tag{66}$$

$$\widehat{W}^{j+1} \stackrel{(59)}{=} \left( \widehat{W}^j + \widehat{\Delta}^j \right)_{:,B+1:}, \tag{67}$$

where $\widehat{\Delta}^j \in \mathbb{R}^{c \times (b - jB)}$ is an update matrix, each line of which is constructed by the rule (60).

In Equation (65), we construct the global residual mask (shown in blue in Figure 7) containing $pcb - \sum_{i=1}^{j} \|M^i\|_F^2$ elements designated for pruning. Here, $pcb$ represents the total number of parameters to be removed, and $\sum_{i=1}^{j} \|M^i\|_F^2$ denotes the cumulative number of parameters pruned in the previous $j$ iterations. Subsequently, in Equation (66), we create a local block mask (shown in red in Figure 7) by selecting only the first $B$ columns from the global residual mask $\widehat{M}^j$. Finally, in Equation (67), we apply the update rule to the weights to calculate $\widehat{W}^{j+1}$, enabling the pruning process to proceed to the next iteration.

Thanos dynamically constructs the global residual mask $\widehat{M}^j$ based on the number of elements already pruned and the target sparsity level. Consequently, this algorithm is not restricted to local block-wise or row-wise sparsity patterns, enabling effective handling of weight adjustments throughout pruning.

A visual representation of this process is provided in Figure 7.

### H.4.2 STRUCTURED SPARSITY

Structured sparsity can be more useful for practical usage because it does not require special cores for acceleration. In this type of sparsity, the entire column is removed, so the weight matrix $W$ can be made smaller without the need for additional storage of indices.

In case when we remove weights by eliminating the entire column, we do not need to solve the equation (56) for every single row since all indices $q_1, \cdots, q_s$ will be the same. For structured pruning, we can apply permutation of the columns of $W$ to make $q_1 = 1, \cdots, q_s = s$, so we only would need to remove the first $s$ columns of the weights matrix $W$. After all of the required transformations we need to perform the inverse permutation to get the correct weight matrix with all columns in the right order. For structured sparsity, the equation (56) can be written in the following form:

$$\widehat{\Delta}_{k:} = -W_{:,1:s} \left( H_{1:s,1:s}^{-1} \right)^{-1} H_{1:s,:}^{-1}. \tag{68}$$

In our experiments, we found that structured pruning is complicated because of the outlier weights: some weights in the column seems to be too important to be effortlessly removed.

### H.4.3 OUTLIER ROWS (DEFINITION OF $\alpha$)

Structured pruning damages every row of $W$, so it might be beneficial not to touch some rows. Let $h_i$, $i \in \{1, \cdots, c\}$ to be the loss (17) induced by removal of the $i^{\text{th}}$ row of the weight matrix $W$, then $v_j$ can be written as

$$h_i := \|W_i X\|_2^2. \tag{69}$$

Let us denote the percent of outliers as $\alpha \in [0, 1)$, where $\alpha = 0$ means we prune all rows, $\alpha = 0.5$ means $50\%$ of rows are outlier-rows, we will not prune them, $\alpha = 1$ means that we perceive all rows as outliers, so nothing will be pruned. The number of outlier rows is $\lceil \alpha c \rceil$.

Note that if your goal is to achieve a fixed level of sparsity $p$, then you have to remove $s = \lceil \frac{pb}{1-\alpha} \rceil$ columns. As expected, with increasing $\alpha$, we have to prune more columns.

We choose columns for removal by computing the loss induced by every column and selecting those with the smallest value. Let $v_j$, $j \in \{1, \cdots, b\}$ to be the partial loss (17) induced by removal of the $j^{\text{th}}$ column of the weight matrix $W$, then $v_j$ can be written as

$$v_j := \|W_{1:c-\lceil \alpha c \rceil, j} \otimes X_j\|_F^2. \tag{70}$$

In total, we perform two permutations – horizontal (permute columns to move weights for removal in the beginning) and vertical (permute rows to move outliers in the end). After the pruning procedure, we have to apply the inverse permutations to obtain the correct weight matrix with all columns and rows in the right order.

### H.4.4   COLUMN AND ROW PERMUTATIONS

To facilitate structured pruning while preserving row-wise and column-wise order after pruning, we employ permutation matrices to reorder the rows and columns of the weight matrix $W$. In this section, we describe how these permutation matrices are defined and demonstrate their use within the pruning procedure.

Let $W \in \mathbb{R}^{c \times b}$ be a weight matrix whose rows and columns we wish to reorder according to certain importance metrics. We denote the loss (or importance) of each column by $\{v_j\}_{j=1}^b$ and the loss of each row by $\{h_i\}_{i=1}^c$. We aim to reorder the columns so that the columns with the smallest $v_j$ appear first, and similarly reorder the rows so that the rows with the smallest $h_i$ appear first (or last, depending on the pruning criterion).

A permutation matrix is a square binary matrix in which each row and each column contains exactly one entry equal to $1$, and all other entries are $0$. Such matrices are orthogonal, so if $P$ is a permutation matrix, then $P^{-1} = P^T$. We will use two permutation matrices:

1. $P \in \{0,1\}^{b \times b}$ for permuting the columns of $W$.
2. $Q \in \{0,1\}^{c \times c}$ for permuting the rows of $W$.

Given the set of column-loss values $\{v_j\}_{j=1}^b$, we define a permutation $\sigma^v$ on the index set $\{1, \ldots, b\}$ by sorting the columns in ascending order of their losses:

$$v_{\sigma^v(1)} \leq v_{\sigma^v(2)} \leq \cdots \leq v_{\sigma^v(b)}.$$

That is, $\sigma^v(k)$ is the index of the $k$-th smallest column loss. We then construct the matrix $P \in \{0,1\}^{b \times b}$ by setting

$$P_{j,\sigma^v(j)} = 1 \quad \text{for all } j = 1, \ldots, b,$$

and zero elsewhere. Intuitively, the $j$-th row of $P$ selects the $\sigma^v(j)$-th column in the new ordering.

Once $P$ is defined, the column-permuted version of $W$ is obtained by right-multiplication:

$$W_{\text{perm}} = W P.$$

The inverse operation simply uses the transpose $P^\top$, which restores the original column ordering:

$$W = W_{\text{perm}} P^\top.$$

Analogously, for the row-loss values $\{h_i\}_{i=1}^c$, define a permutation $\sigma^h$ on $\{1, \ldots, c\}$ by sorting rows in ascending order:

$$h_{\sigma^h(1)} \leq h_{\sigma^h(2)} \leq \cdots \leq h_{\sigma^h(c)}.$$

We then construct $Q \in \{0,1\}^{c \times c}$ by setting

$$Q_{\sigma^h(i),\, i} = 1 \quad \text{for all } i = 1, \ldots, c,$$

with all other entries of $Q$ being zero. The $i$-th column of $Q$ thus selects row $\sigma^h(i)$.

To reorder the rows of $W$, we multiply on the left by $Q$:

$$W_{\text{perm}} = Q W.$$

Similarly, to invert this operation and retrieve the original row ordering, we use $Q^\top$:

$$W = Q^\top W_{\text{perm}}.$$

If one wishes to permute rows and columns simultaneously according to $\sigma^h$ and $\sigma^v$, one can write

$$W_{\text{perm}} = Q W P,$$

where $Q \in \{0,1\}^{c \times c}$ and $P \in \{0,1\}^{b \times b}$ are constructed as described above. The inverse operation recovers the original matrix $W$:

$$W = Q^\top W_{\text{perm}} P^\top.$$

---

**Algorithm 7** Thanos pruning algorithm (Structured)

---

1: **Initialization:** Input matrix $X \in \mathbb{R}^{b \times a}$, weight matrix $W \in \mathbb{R}^{c \times b}$, percent of outlier rows $\alpha \in [0, 1)$, sparsity ratio $p \in [0, 1)$.
2: $s \leftarrow \lceil \frac{pb}{1-\alpha} \rceil$   *// number of columns to remove*
3: $H \leftarrow 2XX^\top$   *// Compute the Hessian*
4: Compute $\{h_i\}_{i=1}^c$ by (10)
5: $Q \leftarrow Q(W, \{h_i\}_{i=1}^c)$   *// Compute rows permutation*
6: $W' \leftarrow QW$   *// Permute rows*
7: Compute $\{v_j\}_{j=1}^b$ by (11)
8: $P \leftarrow P(W'_{1:c-\lceil \alpha c \rceil}, \{v_j\}_{j=1}^b)$   *// Compute columns permutation*
9: $W \leftarrow PW'$   *// Permute columns*
10: $W_{:,1:s} \overset{(8)}{\leftarrow} W_{:,1:s} - W_{:,1:s} \left( H_{1:s,1:s}^{-1} \right)^{-1} H_{1:s,:}^{-1}$   *// prune the matrix*
11: $W \leftarrow Q^\top W P^\top$   *// Perform the inverse permutations*

---

### H.4.5 STRUCTURED $n\!:\!m$ SPARSITY

The Thanos algorithm can be readily extended to support semi-structured pruning patterns, such as the widely-used $n : m$ sparsity format (Zhou et al., 2021; Hubara et al., 2021a; Mishra et al., 2021). This format, which enforces that each block of $m$ consecutive weights contains exactly $n$ zeros, provides computational speedups, especially on NVIDIA Ampere GPUs with its optimized 2:4 implementation (Lin et al., 2023; NVIDIA Corporation, 2020).

For this $n : m$ structure, there is no need to construct the global residual mask. Instead, the focus shifts to creating a local block-structured mask, following the same methodology used in SparseGPT. The structured version of the Thanos pruning algorithm is presented in Algorithm 7.

The computational efficiency of Thanos is enhanced in this semi-structured setting because each row has a uniform number of weights designated for removal. This uniformity eliminates the need for matrix padding, as the systems of linear equations now have consistent dimensions across all rows. Consequently, the pruning process becomes more streamlined, further reducing computational overhead and allowing for more efficient matrix operations within the Thanos algorithm.

### H.4.6 HOW TO GET INDICES OF WEIGHTS FOR REMOVAL

As discussed previously in Section H.1, we require the indices $1 \leq q_1 < \cdots < q_s \leq b$ of the weights designated for removal. In Section H.4, we outlined the method for constructing the local pruning mask; now, our goal is to extract the indices for removal from this mask. To facilitate this, we define a mapping $\phi : \{0,1\}^B \to \mathbb{N}^s$ that transforms each vector $h \in \{0,1\}^B$ into a vector of natural indices of its non-zero elements:

$$\phi(h) := \text{ vector of indices of non-zero elements from } h. \tag{71}$$

For instance, applying the mapping $\phi$ to the vector $h = (1, 0, 0, 1, 1)$ yields $\phi(h) = (1, 4, 5)$. Similarly, for $h = (0, 0, 1, 1, 0)$, we obtain $\phi(h) = (3, 4)$, and so forth.

---

**Algorithm 8** Thanos pruning algorithm for semi-structured $n\!:\!m$ sparsity

---

1: **Initialization:** Input matrix $X \in \mathbb{R}^{b \times a}$, weight matrix $W \in \mathbb{R}^{c \times b}$, block size $B$, percent of outlier rows $\alpha \in [0, 1)$, $0 < n < m \leq b$.
2: $H \leftarrow 2XX^\top$   *// Compute the Hessian*
3: Compute $\{h_i\}_{i=1}^c$ by (10)
4: $Q \leftarrow Q(W, \{h_i\}_{i=1}^c)$   *// Compute rows permutation*
5: $W \leftarrow QW$   *// Permute rows*
6: **for** $j_1 = 1, B, 2B, \cdots, \lfloor \frac{b}{B} \rfloor B$ **do**
7:     $j_2 \leftarrow \min\{b, j_1 + B\}$
8:     $M \leftarrow 0 \in \mathbb{R}^{c \times B}$
9:     **for** $j = 0, m, 2m \cdots, \lfloor \frac{B}{m} \rfloor m$ **do**
10:         $M_{:,j:j+m} \leftarrow$ mask of $n$ weights $W_{kq}$ from the matrix $W_{:,j_1+j:j_1+j+m}$ with smallest$|W_{kq}|\|X_{q:}\|_2$.
11:     **end for**
12:     **for** $i = 1, \cdots, \lceil c(1-\alpha) \rceil$ **do**
13:         $w \leftarrow W_{i,j_1:b}$   *// select one row of weights*
14:         $q \overset{(4.5)}{\leftarrow} \phi(M_{i:}) \in \mathbb{N}^s$   *// find indexes of weights for removal in the row*
15:         $R \overset{(7)}{\leftarrow} \left( (H_{q_1:}^{-1})^\top \cdots (H_{q_s:}^{-1})^\top \right)^\top$
16:         $\widehat{R} \overset{(7)}{\leftarrow} (R_{:q_1} \cdots R_{:q_s})$
17:         $u \overset{(7)}{\leftarrow} (w_{q_1} \cdots w_{q_s})$
18:         $W_{i,j_1:b} \overset{(8)}{\leftarrow} w - u\widehat{R}^{-1}R$   *// update the selected row*
19:     **end for**
20:     $H \leftarrow 2(XX^\top)_{j_2:b,j_2:b}$   *// update the Hessian to consider only unseen weights*
21: **end for**
22: $W \leftarrow Q^\top W$   *// Perform the inverse rows permutation*

---

With this mapping, we can retrieve the indices for removal in each row of the mask $M$ as follows:

$$(q_1, \cdots, q_s) = \phi(M_{i:}),$$

where $M_{i:}$ represents elements from the $i^{\text{th}}$ row of the mask $M$. The mapping $\phi$ allows us to systematically identify the indices of the weights to be pruned within each specified row of the mask.

## H.5 THE MAIN ALGORITHM

## H.6 COMPLEXITY

In this section we estimate the total complexity of Thanos for pruning of one LLM layer. We have to make the following steps to apply Thanos:

- Compute $H = 2XX^\top$ in $\mathcal{O}(ab^2)$ and its inverse for every block of pruning in $\mathcal{O}\left(\frac{b^4}{B}\right)$

- Calculate pruning metric $S_{kj}^{\text{OBD}}$ for every block and find smallest values in $\mathcal{O}\left(\frac{cb^2}{B}\log(cb)\right)$

- Calculate the update rule and update weights for every block in $\mathcal{O}\left(cbB^2 + \frac{cb^2}{B}\right)$

Total complexity is

$$T_{\text{Thanos}} = \mathcal{O}\left(ab^2 + \frac{b^4}{B} + \frac{cb^2}{B}\log(cb) + cbB^2\right) \tag{72}$$

Table 4 presents a comparison of the computational complexities of all methods under consideration.

---

**Algorithm 9** Thanos pruning algorithm (Unstructured)

---

1: **Initialization:** Input matrix $X \in \mathbb{R}^{b \times a}$, weight matrix $W \in \mathbb{R}^{c \times b}$, block size $B$, sparsity ratio $p \in [0, 1)$.
2: $r \leftarrow pcb$    *// number of parameters to remove*
3: $H \leftarrow 2XX^{\top}$    *// Compute the Hessian*
4: **for** $j_1 = 1, B, 2B, \cdots, \lfloor \frac{b}{B} \rfloor B$ **do**
5:     $j_2 \leftarrow \min\{b, j_1 + B\}$
6:     $\widehat{M} \overset{(45)}{\leftarrow} \psi_X(W_{:,j_1:b}, r)$    *// global residual mask*
7:     $M \leftarrow \widehat{M}_{:,1:B}$    *// local block mask*
8:     $r \leftarrow r - \|M\|_F^2$    *// update number of parameters to remove*
9:     **for** $i = 1, \cdots, c$ **do**
10:       $w \leftarrow W_{i,j_1:b}$    *// select one row of weights*
11:       $q \overset{(71)}{\leftarrow} \phi(M_{i:}) \in \mathbb{N}^s$    *// find indexes of weights for removal in the row*
12:       $R \overset{(52)}{\leftarrow} \left((H_{q_1:}^{-1})^{\top} \cdots (H_{q_s:}^{-1})^{\top}\right)^{\top}$
13:       $\widehat{R} \overset{(54)}{\leftarrow} (R_{:q_1} \cdots R_{:q_s})$
14:       $u \overset{(7)}{\leftarrow} (w_{q_1} \cdots w_{q_s})$
15:       $W_{i,j_1:b} \overset{(56)}{\leftarrow} w - u\widehat{R}^{-1}R$    *// update the selected row*
16:     **end for**
17:     $H \leftarrow 2(XX^{\top})_{j_2:b,j_2:b}$    *// update the Hessian to consider only unseen weights*
18: **end for**

---

# I IMPLEMENTATION DETAILS

## I.1 EFFICIENT BATCHED OPERATIONS

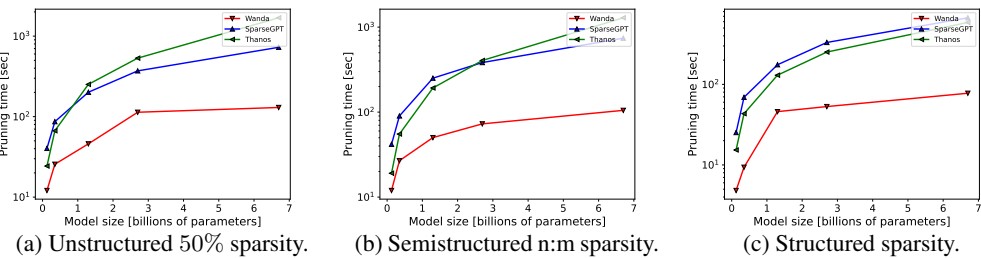

(a) Unstructured 50% sparsity.     (b) Semistructured n:m sparsity.     (c) Structured sparsity.

Figure 8: Comparison of pruning time for different pruning methods across different model's sizes for OPT (Zhang et al., 2022) family models. Thanos is more efficient than SparseGPT especially in case of structured sparsity.

In developing the Thanos pruning algorithm, several techniques were employed to ensure computational efficiency and scalability. The naive version of the Algorithm 9 used nested loops to demonstrate basic functionality on small models. However, this approach proved inefficient as model sizes increased. To optimize performance, broadcasting operations in NumPy (Harris et al., 2020) and PyTorch (Paszke et al., 2019) replaced these loops, leveraging underlying C++ optimizations to significantly reduce computational overhead.

A core challenge in Thanos was pruning weights across each row, with different rows requiring varied amounts of weights that should be removed. This necessitated solving linear systems of equations of varying sizes per row, which hindered the direct use of batched linear solvers in PyTorch that assume uniform dimensions. To overcome this, matrices were expanded to match the maximum row size by adding padding where necessary.

Specifically, we considered this padding approach as follows: given a local block mask $M \in \{0, 1\}^{c \times B}$, we first determined the maximum number of weights to be removed across all rows,

defined as $r_{\max} := \max_{i \in \{1, \cdots, c\}} \|M_{i:}\|_F^2$. For each row being pruned, we used the mapping $\phi$, as defined in Equation (71), to obtain the indices of weights designated for removal. To ensure uniformity, we padded each vector of indices to have a consistent size of $r_{\max}$ by filling it with ones as needed. For example, if $r_{\max} = 5$ and $\phi(M_{i:}) = (1, 2, 5)$, the padded vector of indices would become $(1, 2, 5, 1, 1)$.

The next step is to construct the modified vector $u$, defined in (7):

$$u' := (w_{q_1} \cdots w_{q_s} \ 0 \ \cdots \ 0) = \begin{pmatrix} u & 0 \in \mathbb{R}^{1 \times (r_{\max} - s)} \end{pmatrix} \in \mathbb{R}^{1 \times r_{\max}}. \tag{73}$$

Next, we modify the matrix $\widehat{R}$, as defined in Equation (54), to ensure it has consistent dimensions across all rows of $W$:

$$\widehat{R}' := \begin{pmatrix} & & & 0 & \cdots & 0 \\ & \widehat{R} \in \mathbb{R}^{s \times s} & & \vdots & \ddots & \vdots \\ & & & 0 & \cdots & 0 \\ 0 & \cdots & 0 & 1 & & 0 \\ \vdots & \ddots & \vdots & & \ddots & \\ 0 & \cdots & 0 & 0 & & 1 \end{pmatrix} \in \mathbb{R}^{r_{\max} \times r_{\max}}. \tag{74}$$

Finally, we construct the modified version of system of linear equations from (53):

$$\widehat{\lambda}' \widehat{R}' = u', \tag{75}$$

where $\widehat{\lambda}' \in \mathbb{R}^{1 \times r_{\max}}$. It is straightforward to observe that the modified system of linear equations, as defined in Equation (75), will yield $s$ non-trivial solutions. By construction of the matrix $\widehat{R}'$ and the vector $u'$, all solutions

$$\widehat{\lambda}'_j = 0 \quad \text{for } j > s.$$

This adjustment enabled the use of batched solvers, thus enhancing computational efficiency while maintaining flexibility across non-uniform rows. For models smaller than 1 billion parameters, Thanos demonstrates greater speed compared to SparseGPT. This efficiency arises because, unlike SparseGPT, which prunes parameters individually, Thanos adopts a block-wise pruning approach. By processing parameters in blocks rather than one at a time, Thanos significantly reduces computation time, making it particularly effective for smaller-scale problems within this parameter range (Figure 8a).

## I.2 GPU MEMORY LIMITATIONS

The systems of linear equations in Equation (75) are solved simultaneously for all rows using batched linear solvers in PyTorch. However, as the matrix sizes increase, this operation can exceed the available GPU memory, even on high-memory GPUs like the NVIDIA A100 with 40 or 80 GB of memory. These GPUs may struggle to accommodate such systems for all rows simultaneously, even for relatively small models such as OPT-1.3B (Zhang et al., 2022).

To address this, the Thanos algorithm introduced a further subdivision within the pruning process. Instead of pruning an entire column across all rows simultaneously, each column is divided into manageable vertical blocks, ensuring each block of linear equations fits within the GPU's memory constraints.

By processing these blocks sequentially, we solve the system of equations for all rows within each block, then proceed to the next vertical block within the column. This systematic segmentation not only makes it feasible to fit larger matrices onto the GPU but also maintains efficient pruning operations without excessive memory consumption. As a result, this technique enables Thanos to handle large batch of linear system of equations on GPUs efficiently, despite memory limitations.

## I.3 ACCELERATED STRUCTURED $n:m$ SPARSITY

While Thanos was originally designed for unstructured sparsity, the algorithm also extends to structured sparsity (e.g., 2:4 or 4:8 patterns), where a uniform number of weights are pruned per row.

This uniformity eliminates the need for padding adjustments described in Section I.1, since now every row will have the same number of weights for removal.

When structured sparsity is applied, Thanos can be even more efficient and faster than SparseGPT for model sizes up to 3 billion parameters (Figure 8). This efficiency gain is due to Thanos block-wise pruning strategy, which aligns well with structured sparsity patterns. By pruning multiple weights within structured blocks, Thanos minimizes computational overhead, making it highly effective for medium-sized models and allowing it to outperform SparseGPT in these settings.

