# OpenReview forum: "Thanos: A Block-wise Pruning Algorithm for Efficient Large Language Model Compression"
_ICLR.cc/2026/Conference — Submitted to ICLR 2026_

### Official Review · Reviewer_kKnK · 2025-10-17

**Soundness:** 2
**Presentation:** 1
**Contribution:** 1
**Rating:** 2
**Confidence:** 3

**Summary:**

This paper proposes Thanos, a block-wise pruning algorithm for large language models. It introduces adaptive masks and joint weight updates to better capture the impact of pruning multiple parameters, supporting unstructured, structured, and n:m sparsity patterns. Experiments on LLaMA and OPT show that Thanos outperforms baselines like SparseGPT, Wanda, and magnitude pruning in both perplexity and zero-shot accuracy, especially under structured sparsity.

**Strengths:**

The paper addresses an important problem in LLM compression and proposes a pruning method that is relatively straightforward to implement. It covers both unstructured and structured sparsity settings, including n:m formats, which makes the method more practically relevant. The experimental section evaluates Thanos across multiple model scales and tasks, ensuring a fair comparison with widely used baselines.

**Weaknesses:**

1. The paper is poorly organized; for example, the conclusion is placed in the appendix instead of the main body, and the overall structure lacks clarity.
2. A large portion of the paper is devoted to re-describing prior methods, which seems unnecessary and distracts from the core contribution.
3. The main novelty—an extension of SparseGPT with multi-element joint pruning updates—appears incremental and insufficient to support a full paper.
4. The experimental evaluation is incomplete: it does not report model size reduction or inference speedup, and the improvements over Wanda are marginal.
5. There is no ablation study to disentangle the contribution of each proposed component.
6. The choice of models in Table 1 is unclear: the paper refers to “LLaMA-2-1.1B” and “LLaMA-3-1B,” which do not correspond to official Meta releases, leaving ambiguity about what models are actually used.

**Questions:**

Please refer to the weakness section.

---

### Official Review · Reviewer_25Jh · 2025-10-22

**Soundness:** 2
**Presentation:** 3
**Contribution:** 2
**Rating:** 4
**Confidence:** 3

**Summary:**

The paper introduces Thanos, a post-training pruning algorithm aimed at shrinking large language models (LLMs) without retraining. Thanos works by partitioning every linear layer into manageable column-wise blocks and, for each block, jointly selecting multiple weights to drop and analytically re-optimising the surviving weights. This is achieved by adaptive pruning mask strategy and update multiple weights simultaneously. Moreover, they devise a structured variant that permutes rows and columns so that whole columns can be excised while optionally preserving a user-defined fraction of "outlier" rows.

**Strengths:**

1. The proposed Thanos algorithm introduces a block-wise pruning algorithm with joint weight updates. This improves upon prior work (SparseGPT) that only prunes one weight per row at a time. The paper also offers well-grounded theoretical derivations and practical heuristics, such as adaptive mask updates and outlier rows compatibility method, retaining model generalization capabilities.
2. The Thanos supports unstructured, structured, and semi-structured (n:m) sparsity patterns. The adaptability to formats like 2:4 allows it to take advantage of hardware acceleration.
3. Extensive empirical evaluations across different models show that Thanos outperforms existing methods (Magnitude, Wanda, SparseGPT) on both perplexity and zero-shot tasks.

**Weaknesses:**

1. While the paper explores the effect of mask strategy and joint weight updates, it lacks a thorough ablation study to isolate the individual contributions of different algorithmic components. Such analysis would strengthen the empirical claims.
2. The strategy of retaining outlier rows may hinder strict structural pruning. For instance, it becomes unclear whether entire columns can always be pruned, which could compromise compatibility with hardware acceleration schemes relying on full-column removal. The paper does not sufficiently clarify how Thanos maintains structured sparsity under these conditions.
3. The evaluation primarily focuses on perplexity and zero-shot accuracy, but omits key practical metrics such as pruning time and inference-time memory usage. These factors are crucial when choosing among pruning methods for deployment. Including such comparisons would have significantly enhanced the practical value of the results.

**Questions:**

Please refer to the "Weakness" section.

---

### Official Review · Reviewer_qK9D · 2025-10-27

**Soundness:** 3
**Presentation:** 3
**Contribution:** 3
**Rating:** 8
**Confidence:** 2

**Summary:**

This paper focuses on the pruning of the transformer block to remove the redundancy but preserve the accuracy. Authors focuses on the post-training pruning and introduces a method named Thanos. Thanos dynamically constructs the global residual mask based on the number of elements already pruned and the final target. It also take a hyperparam alpha to take the outlier row into consideration for preserving the useful outlier row. Authors have conducted extensive experiemnts on multiple matrices, including perplexity analysis, zero-shot performance and also ablation with different block size, where the proposed Thanos show state-of-the-art performance for all of these metrics.

**Strengths:**

+ The proposed model is well described and show very promising results compared with other existing state-of-the-art methods.

+ Authors also provide extensive analysis for the different datasets in supplementary material, which is showing the proposed method showing a very promising results.

**Weaknesses:**

- If authors are able to provide some analysis in addition to Llama and OPT, it would be great to show the overall generalizability of the proposed model.

- Please consider include part of the results in section E to the main manuscript as these are very important numbers.

**Questions:**

NA

---

### Official Review · Reviewer_fuCo · 2025-10-31

**Soundness:** 2
**Presentation:** 3
**Contribution:** 2
**Rating:** 4
**Confidence:** 4

**Summary:**

This paper proposes Thanos, a block-wise post-training pruning algorithm for large language models. Unlike prior methods such as SparseGPT and Wanda that prune weights independently, Thanos removes multiple weights jointly within each block, leveraging local Hessian information to better preserve model performance. It introduces adaptive masking based on weight–input importance and supports unstructured, structured, and semi-structured sparsity, including an outlier row preservation mechanism for stability at high sparsity. Experiments on OPT and LLaMA series show that Thanos achieves state-of-the-art perplexity and zero-shot accuracy under both unstructured and structured pruning, while maintaining computational efficiency suitable for hardware acceleration.

**Strengths:**

The paper is generally well-organized. The proposed block-wise pruning approach is thoughtfully designed, offering a reasonable trade-off between pruning accuracy and computational efficiency. The inclusion of outlier-row preservation and support for structured and semi-structured sparsity makes the method adaptable to practical deployment. The experimental section is fairly comprehensive, and the presentation of algorithms and results is clear and easy to follow.

**Weaknesses:**

The evaluation is somewhat narrow, focusing mainly on OPT and LLaMA, leaving uncertainty about generalization to newer models. The paper lacks detailed ablation or sensitivity studies on key hyperparameters such as block size and α. Reported efficiency gains are not well supported by runtime or memory analyses.

**Questions:**

1. Could the authors evaluate Thanos on more recent or diverse model families (e.g., Qwen3) to better demonstrate its generalization across architectures?
2. Can the authors provide more detailed ablation or sensitivity analyses on key hyperparameters such as block size (B) and outlier ratio (α), especially across different sparsity levels or model scales?
3. The paper claims improved efficiency. Could the authors include quantitative comparisons (e.g., runtime or throughput) to substantiate these efficiency gains?

---

### Meta-Review · Area_Chair_nACM · 2026-01-07

**Summary:**

This manuscript proposes Thanos, a post-training pruning method for LLMs. The method proposes joint pruning of weights within each block and re-optimizes the remaining weights using an analytical formulation. It can be applied to structured, semi-structured, and unstructured pruning settings, and achieves competitive accuracy results.

The reviewers show clearly divergent scores on this paper. The major concern is the lack of system-level inference evaluation, including real speedup, memory/storage savings, and other practically relevant metrics.

**Reviewer Concerns:**

(1) Some reviewers questioned the novelty of the method and how much each component actually contributes. The rebuttal does not include clear ablation studies that isolate the effect of individual design choices. Although the authors added more experiments on hyperparameters and settings, these do not fully resolve the concern about methodological contribution.

(2) More than one reviewer raised concerns about real-world efficiency, including actual inference acceleration, memory usage, and calibration or pruning time cost. The authors did not provide concrete measurements or quantitative analysis in the rebuttal. As a result, this concern is still not addressed.

(3) Reviewers also questioned whether the evaluated models are outdated. The experiments mainly focus on OPT and LLaMA. The authors did not explain how the method would apply to more recent model families, nor did they provide experiments on newer architectures. In addition, the evaluation tasks and datasets are relatively simple, which makes the empirical evidence less convincing.

**Reviewer Scores:**

The reviewer scores are highly polarized, ranging from 2 to 8. Since the authors did not provide new experimental evidence that directly addresses the main concerns, this disagreement remains after rebuttal. Reviewer qK9D gives a high score (8: accept), but the review itself is rather general and superficial, and does not seriously engage with the core issues raised by other reviewers. The other reviewers provide more detailed and critical feedback, leading to a borderline overall assessment.

---

### Decision · Program_Chairs · 2026-01-26

Reject